



# Spatiotemporal scales of mode water transformation in the Sea of Oman

Estel Font[1], Esther Portela[2], Sebastiaan Swart[1,3], Mauro Pinto-Juica[1], Bastien Y. Queste[1]

[1]Department of Marine Sciences, University of Gothenburg, Gothenburg, Sweden
[2]Laboratoire d'Océanographie Physique et Spatiale, University of Brest, CNRS, IRD, Ifremer, Plouzané, France
[3]Department of Oceanography, University of Cape Town, Rondebosch, South Africa

*Correspondence to*: Estel Font (estel.font.felez@gu.se)

**Abstract.** In the Sea of Oman, mode water forms at the surface and is trapped under a warm stratified layer in summer. This
capped and well-mixed oxygenated layer decouples the oxygen minimum zone from ocean surface processes and provides a
space for remineralisation, reducing oxygen demand in the deeper oxygen minimum zone. Several physical processes, from
isopycnal and diapycnal mixing to advection, transform mode water and change its properties. Using monthly climatologies
derived from profiling floats and high-resolution underwater glider observations, we perform a volume budget analysis to
investigate the mechanisms driving mode water volume change in the Sea of Oman from monthly to 3-day temporal scales.
Isopycnal and diapycnal water-mass transformations are estimated in a density-spice framework. Mode water predominantly
transforms along isopycnals, yet strong but transient diapycnal transformation occurs at shorter timescales. Moreover, fluxes
between the mode water layer and its surroundings are highly sensitive to the presence of mesoscale eddies. Across eddies,
diapycnal and isopycnal transformations intensify by 61% and 45% respectively, compared to non-eddy conditions,
indicating that eddies are drivers of both lateral and vertical water mass exchanges. This study provides a new
methodological approach to understanding water mass transformation using high-resolution underwater gliders, and shows
that this water mass transformation framework can be used at higher resolution than traditional climatological products or
models. By comparing monthly climatological products to the high-resolution glider data, we estimate that the climatological
estimates are outside of the high-resolution glider mean ± standard error 40% of the time for diapycnal and 60% of the time
for isopycnal transformation. These results highlight the intense variability occurring at small scales and can serve to inform
future estimates of water mass transformation uncertainty from coarser products.

## 1 Introduction

Mode waters are vertically homogeneous waters formed in the surface mixed layer and then subducted into the ocean interior
(Hanawa and Talley, 2001). Surface water properties and tracers (e.g, heat, oxygen, organic matter) are trapped below the
mixed layer during mode water formation. The seasonal variations of the mixed layer result in alternating mixing and
stratification periods, trapping mode waters in the ocean subsurface, and favoring heat uptake, carbon sequestration, and



oxygen ventilation (Lacour et al., 2023; Li et al., 2023; Portela et al., 2020a). Thus, mode water plays a crucial role as a pathway connecting the surface and subsurface ocean, influencing the distribution and transport of oxygen, heat, and remineralized organic matter.

In the Sea of Oman, mode water forms when springtime warming and weak turbulent fluxes stratify the surface layer and cap the deep surface mixed layer formed during the winter monsoon (Font et al., 2022, 2025; Senafi et al., 2019). The fate of mode water determines whether subducted surface properties are retained and transported into the ocean interior or mixed back into the surface mixed layer. This fate is governed by a variety of complex physical processes and instabilities that act at different spatial and temporal scales, resulting in diapycnal and isopycnal mixing and advection of mode water properties.
Thus, these processes collectively regulate the transformation, ventilation, and residence time of mode water, thereby controlling its capacity to act as an oxygen reservoir (Kalvelage et al., 2015), or as a remineralization buffer layer (Weber and Bianchi, 2020).

In the Sea of Oman, mode water volume and residence time are among the largest within the Arabian Sea (Font et al., 2025),
making it a key reservoir of oxygen. Mode water provides significant opportunities for the remineralization of labile sinking organic matter within this layer, reducing biological oxygen demand in the core of the oxygen minimum zone (Weber and Bianchi, 2020). Moreover, mode waters are oxygen-rich waters, which can sustain the oxygen supplied to the upper oxygen minimum zone across the oxycline. It has been shown that, for instance, 50% of the oxygen changes along mode water ventilation pathways are due to respiration within the water mass, the rest being due to mixing with surrounding oxygen-
poorer waters (Jutras et al., 2025). Given their role in ocean heat uptake, biogeochemical cycling, and carbon sequestration, the life cycle of mode water and its changes have significant implications for both regional and global ocean-climate dynamics. Understanding the physical drivers that mix and stratify mode water at different scales and its coupling with biophysical processes is critical. Nevertheless, the timescales over which mode water characteristics and properties change remain poorly constrained, primarily due to limited observations at the spatiotemporal scales required to characterize
subseasonal and small-scale processes and variability.

Based on the method first introduced by Walin (1982) to compute water-mass transformation in thermohaline coordinates, we used a density-spice ($\sigma$-$\tau$) framework (Portela et al., 2020b) to investigate volume changes at timescales, from daily to monthly, of mode water in the Sea of Oman leveraging high spatio-temporal resolution from underwater glider observations
and Argo monthly climatologies. Potential density is the natural coordinate to separate isopycnal and diapycnal fluxes, while spice is added as a second dimension in the volume budget to identify different water masses spreading along isopycnals (Jackett and McDougall, 1985; McDougall and Krzysik, 2015). Spice acts as a tracer of thermohaline structure along isopycnals, enabling the identification of water masses that are indistinguishable in density but differ in origin or transformation history. The water mass transformation framework has been widely employed to quantify how water masses





change properties over seasonal to annual timescales, often using Argo float data or numerical models (Badin et al., 2013;
Evans et al., 2014; Nurser et al., 1999; Portela et al., 2020b). These studies reveal gradual seasonal transformation driven by
surface fluxes and large-scale circulation. However, evidence highlights the significant contribution of mesoscale eddies and
other transient processes to shorter-term variability and transformation events (e.g., Liu et al., 2013; Shi et al., 2018; Trott et
al., 2019; Thoppil, 2024; Xu et al., 2016). Our approach and a high-resolution dataset allow us to capture the episodic and

localized transformations often missed by coarser-resolution climatologies and models, to resolve short-term drivers of
isopycnal and diapycnal change.

Specifically, this study addresses the following key questions: (1) How does mode water change across different timescales,
from days to seasons? (2) Which physical mechanisms predominantly drive mode water transformations in the Sea of

Oman? By combining glider data with climatological products, we provide a quantitative assessment of water mass
transformation dynamics of this layer with important biogeochemical regional implications.

## 2. Data and methods

### 2.1 Ocean glider observations and Argo climatology

We utilized four months of glider observations sampling across the continental shelf in the southern Sea of Oman. The

Seaglider carried out repeated sampling along an 80-km transect between 23.65°N, 58.65°E and 24.25°N, 59°E in 2015
(Figure 1a). Observations were projected onto a straight across-shelf transect (Figure 1a, blue line) and median-binned onto a
0.5 m (vertical) and 2 km (horizontal) grid. The origin of coordinates was taken as the 300 m isobath. Binned sections were
interpolated vertically, then horizontally to fill minor gaps (1 m vertically and 4 km horizontally), and smoothed horizontally
with a 6 km running mean. Mixed layer depth was defined using a density threshold of 0.125 kg m$^{-3}$ relative to the surface

(Font et al., 2022). Spice ($\tau$; kg m$^{-3}$) was computed from conservative temperature and absolute salinity following the TEOS-
10 routines (McDougall and Barker, 2011). Spice is interpreted as a measure of thermohaline variability along isopycnals,
reflecting isopycnally-compensated temperature and salinity changes (McDougall and Krzysik, 2015). Eddy kinetic energy
(EKE) was estimated from two independent sources: (1) the anomalies of the dive-averaged currents derived from the glider
flight model (Frajka-Williams et al., 2011; Queste et al., 2018a) and (2) from sea surface height-derived surface geostrophic

velocities from satellite during the glider-sampling period. Mesoscale eddies in the region were identified using the
ToEddies dataset, which applies an eddy detection algorithm to satellite-derived Absolute Dynamic Topography (ADT)
fields (Laxenaire et al., 2024). The dataset provides daily information on eddy properties, including polarity, center location,
and spatial extent (Laxenaire et al., 2024; Ioannou et al., 2024).

In addition, we used a total of 3565 Argo profiles of temperature, salinity, and pressure between 2000 and 2023 in the Sea of
Oman (22.5-26 °N, 56-61°E, Figure 1a). Only profiles flagged as "good" by the CORIOLIS Data Centre were used (Gaillard





et al., 2009). Argo profiles were remapped on the across-Sea of Oman transect (Figure 1a, gray dashed line) to construct an across-gulf monthly climatology, with a median binning onto a 2 m (vertical) and 3 km (horizontal) grid. To avoid the influence of shallow profiles on the continental shelf, only profiles with a depth greater than 1000 m and located less than 200 km from the across-Gulf transect were included. Argo float coverage spans the entire domain, with an average density of 28 profiles per 0.25° × 0.25° grid cell and a relatively uniform monthly distribution (Figure 1c).

We defined the mode water core as the layer bounded by the 25 and 25.25 kg m$^{-3}$ isopycnals, which effectively capture the low-stratification and low-potential vorticity typical of mode water. This density-based definition closely aligns with a more traditional identification based on potential vorticity (Feucher et al., 2022; Herraiz-Borreguero and Rintoul, 2011; McCartney, 1982), as demonstrated by a strong correlation between potential vorticity and the potential density threshold method (see Supplementary Figure 1; mode water thickness correlation between methods is r² = 0.96). Using a fixed isopycnal threshold simplifies the interpretation of water mass transformation processes within the σ-τ framework (see Section 2.2), while remaining consistent with dynamically meaningful definitions of mode water.

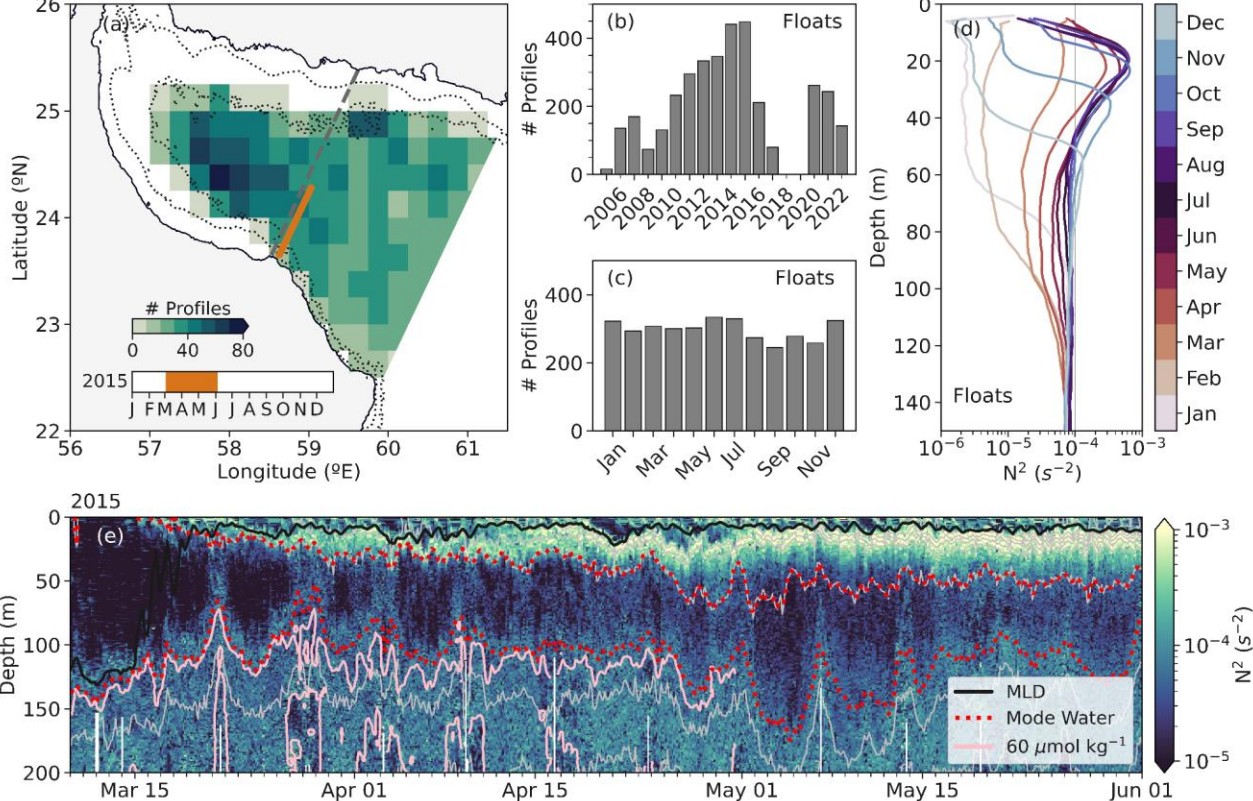

**Figure 1: Data distribution.** (a) Sea of Oman colored by the number of Argo profiles per 0.25°×0.25° geographical grid. Land is shaded gray, the coastline is marked as a solid line, and the contours of the 100 m and 1000 m isobaths are dotted. The dashed gray line is the



transect across the Sea of Oman where Argo profiles are projected to. The glider transect is marked with the orange line, and its duration is
colored in orange in the time bar. (b) Yearly and (c) monthly distribution of the number of Argo profiles in the Sea of Oman. (d) Monthly
climatological stratification (Brunt-Vaisala frequency squared, $N^2$) profiles from Argo data. (e) Stratification time series of the upper 200
m from the glider data with isopycnals in gray. The black line is the mixed layer depth, the upper and lower boundaries of the mode water
layer (25 and 25.25 kg m$^{-3}$ isopycnals, respectively) are marked with dotted red contours, and the 60 μmol kg$^{-1}$ oxygen contour in pink.
Note that after May, no more oxygen data are available due to sensor failure.

## 2.2 Water mass transformation framework

The analyses are performed in $\sigma$-$\tau$ coordinates since: (1) potential density is the natural coordinate to study ocean dynamics
and provides information about the isopycnal and diapycnal nature of the mechanisms driving the volume change; and (2)
spice is defined as the thermohaline changes along isopycnals (following the definition of McDougall and Krzysik, 2015)
and is a proxy for the contrast between water masses spreading along isopycnals (Jackett and McDougall, 1985; McDougall
and Krzysik, 2015).

Based on the constructed Argo monthly climatology and the ocean glider dataset, we performed a volume budget in $\sigma$-$\tau$
coordinates for the interior ocean (i.e., below the mixed layer). The water-mass volume change $dV/dt$ results from the sum of
water-mass formation, $\sum U(\sigma,\tau)$, and the exchange flux across the domain's geographical limits $\Psi$ (Equation 1; Evans et al.,
2014; Portela et al., 2020b). $U$ is the water-mass transformation due to isopycnal and diapycnal mixing fluxes, whereas its
convergence or divergence is represented by the sum ($\sum$) of the interior fluxes across the geographical limits of a given $\sigma$-$\tau$
class (Donners et al. 2005; Walin 1982). The isopycnal and diapycnal transformation rates represent the net mixing of water
masses along and across density surfaces, respectively, resulting in an irreversible property change. $\Psi$ represents the net
exchange across the northern boundary of the domain, defined as the northernmost bin of the glider and climatology transect.
Note that an underestimation of the exchange flux $\Psi$ could result in an apparent overestimation of the local transformation
rate U. However, since both diapycnal and isopycnal transformations are estimated using the same underlying framework,
there is no reason to expect a preferential bias toward either component. In summary, changes in volume within a given $\sigma$-$\tau$
class can be attributed to the convergence or divergence of interior fluxes across $\sigma$ or $\tau$ boundaries, or via exchange fluxes
across the domain boundaries within the same $\sigma$-$\tau$ class. Unlike interior mixing, exchange fluxes do not imply a
transformation of water mass properties.

The method used here to compute water-mass transformation in the $\sigma$-$\tau$ coordinates is based on Portela et al. (2020b) and
Evans et al. (2014). The volume change within a given $\sigma$-$\tau$ class can then be expressed as:

$$\frac{dV}{dt}(\sigma',\tau') = U_\sigma(\sigma',\tau') + U_\tau(\sigma',\tau') + \Psi(\sigma',\tau') \qquad [1]$$

where

$$U_\sigma(\sigma,\tau) = \int \sigma' = \sigma \cdot \boldsymbol{\Pi}(\tau,\tau') \cdot u_\sigma \cdot dA \qquad \text{and} \qquad U_\tau(\sigma,\tau) = \int \tau' = \tau \cdot \Pi(\sigma,\sigma') \cdot u_\tau \cdot dA \qquad [1.1, 1.2]$$



where $\Pi = 1$ if both $\sigma$ and $\tau$ fall within a specific $\sigma$-$\tau$ class defined by $\sigma' = (\sigma \pm \Delta\sigma/2)$ and $\tau' = (\tau \pm \Delta\tau/2)$, and $\Pi = 0$ otherwise. The volume integration was made in $\sigma$-$\tau$ classes with density intervals of $\Delta\sigma$=0.05 kg m$^{-3}$ between 23 and 27.5 kg m$^{-3}$ and spice intervals of $\Delta\tau$=0.1 kg m$^{-3}$ between 4 and 8.5 kg m$^{-3}$. This choice reflects a trade-off between adequately
representing each water mass class and ensuring that the mode water layer is well resolved. Here, $u_\sigma$ and $u_\tau$ are the diapycnal and isopycnal (diaspice, following our assumptions) velocity components, respectively, and $dA$ are the isopycnal and isospice areas covered by the given $\sigma$-$\tau$ class limits, respectively. Equation (1) is solved by means of a least squares regression for the unknown transformation and exchange flow terms using a set of linear equations that link the volume trend to the interior water-mass transformation in $\sigma$-$\tau$ coordinates: dV/dt=Ax, where A is the matrix of coefficients of the linear
equations and **x** is the vector of the resulting diasurface transformations and exchange flux: $\mathbf{x} = (U_\sigma + U_\tau + \Psi)$. The solution **x** was then decomposed into the transformation across spice and density surfaces and the exchange flux across the geographical domain. The detailed methodology can be found in Evans et al. (2014) and Portela et al. (2020b).

The volume of water bounded by a given tracer surface (e.g., here potential density or spice) can change due to mixing
across the tracer surface or due to volume transport into or out of the domain. However, it can also change by air-sea buoyancy fluxes if the tracer surface outcrops at the ocean surface (Evans et al., 2014; 2023; Groeskamp et al., 2019). To assess the role of surface forcing, we identified $\sigma$-$\tau$ classes that intersect the ocean surface and are thus affected by buoyancy fluxes, using ERA5 reanalysis data (Hersbach et al., 2020). However, for the period and $\sigma$-$\tau$ classes of interest for this study, transformation appears unaffected by air-sea buoyancy fluxes (Figure S2). The layers influenced by these fluxes lie well
above the density classes of mode water and are thus not expected to directly influence the observed transformations. Moreover, we exclude a subduction term through the base of the deepest mixed layer. This is justified because we start the water-mass transformation analysis well after mode waters are capped and the $\sigma$-$\tau$ classes lie well below the surface mixed layer under strong stratification (O(10$^{-3}$) s$^{-2}$).

The water-mass transformation framework was applied on 20 ocean glider sections, each 60 km long and extending from the surface to 500 m depth, at the location indicated in orange in Figure 1a. To minimize the influence of shelf processes on the transformation, sections begin 10 km from the 300 m isobath, where depths exceed 500 m. The temporal resolution is approximately 3 days between successive sections. The first transect included in the analysis starts on 25 March, roughly a week (two sections) after restratification, to minimize the influence of potential subduction/exchange with the surface mixed
layer. Moreover, the framework is also applied to the across-gulf monthly climatology to obtain a climatological average of water mass transformations in the region. As restratification of the surface mixed layer typically completes by mid-March (Font et al., 2022), we restrict the analysis of monthly transformations to the period from April onwards. The residual errors for both datasets, computed as $|(dV/dt) - A\mathbf{x}|/|(dV/dt)|$, were on average of order 10$^{-5}$, which is on the higher end of values reported in previous studies (e.g., Portela et al., 2020b).



## 3. Results

### 3.1 Timescales of transformation of mode water

Mode water forms between February and March, and lingers as a homogeneous layer below the seasonal pycnocline (Figure 1d-g), forming a boundary between the mixed layer above and the Arabian Sea oxygen minimum zone below (< 60 μmol kg$^{-1}$) (Figure 1e-g). Climatologically, the low stratification characteristic of mode water in the Sea of Oman is present until June (Figure 1d), but it can persist longer inside eddies (Font et al. 2025; Liu et al., 2013). Over seasonal timescales, the transformation framework reveals a net shift toward mintier and lighter waters, driven by negative isopycnal and diapycnal transformations. Isopycnal transformation (Figure 2b) indicates mixing into colder, fresher classes, while diapycnal transformation (Figure 2c) reflects mixing into lighter classes. Thus, there is water volume formed of lighter and mintier waters (red shading, Figure 2a), but a large destruction predominantly in the lighter and spicier classes of mode water (blue shading, Figure 2a). This fact, alongside a thinning of the mode water layer (ΔThickness=10 m month$^{-1}$; Figure 2d), results in mean mintification and densification of mode water between March and June ($\Delta\tau = $ *-0.03* kg m$^{-3}$ month$^{-1}$, $\Delta\sigma = 0.0013$ kg m$^{-3}$ month$^{-1}$, Figure 2e). On average, the isopycnal transformation dominates (-0.015 ± 0.009 m$^2$ s$^{-1}$), with a magnitude approximately three times larger than the diapycnal transformation (-0.005 ± 0.008 m$^2$ s$^{-1}$) (Figure 2b, c, and f).

The previous analysis was based on monthly climatologies and, therefore, mesoscale and smaller contributions to the transformation are largely averaged out. Over shorter timescales, the high-resolution time series provided by the ocean gliders offers a unique view that highlights the large variability of mode water thickness and depth due to eddy presence and topographical interactions with the shelf break (Figures 1e and 2d). We observe a transition toward mintier waters throughout the study period ($\Delta\tau = $ -0.04 kg m$^{-3}$ month$^{-1}$), albeit with notable temporal variability as indicated by the sign changes in the isopycnal transformation (Figure 2ef). Until early April, waters become spicier (isopycnal tranf. > 0). Then, they shift to mintier conditions (isopycnal transf. < 0), which last until the end of April. After that, waters get spicier again. This behaviour reverses in mid-May, and the period ends with mintier waters (isopycnal tranf. < 0). A similar pattern is observed in the diapycnal transformation. Over the sampled period, waters get slightly lighter ($\Delta\sigma=0.001$ kg m$^{-3}$ month$^{-1}$) but with large variability ($\Delta\sigma = \pm0.02$ kg m$^{-3}$; Figure 2e). Densification occurs until early April (diapycnal tranf. > 0), followed by a period of lightening (diapycnal tranf. < 0), and a shift to densification after mid-May (diapycnal transf. > 0; Figure 2e). Notably, the isopycnal and diapycnal transformations are well correlated through most of the period until they begin to diverge in late April, when the isopycnal transformation changes sign approximately two weeks before the diapycnal transformation (Figure 2f).

Despite the variability, the mean magnitude of the glider-based transformations is of the same order (10$^{-2}$-10$^{-3}$ m$^2$ s$^{-1}$) as the climatological transformation, with twice the contribution from isopycnal to diapycnal transformation, yet larger extremes





for the latter (isopycnal: -0.004 ± 0.027 m$^2$ s$^{-1}$; diapycnal: 0.002 ± 0.029 m$^2$ s$^{-1}$; Figure 2f). We suggest that the high variability in both positive and negative diapycnal transformations is averaged out in the monthly climatological sections.

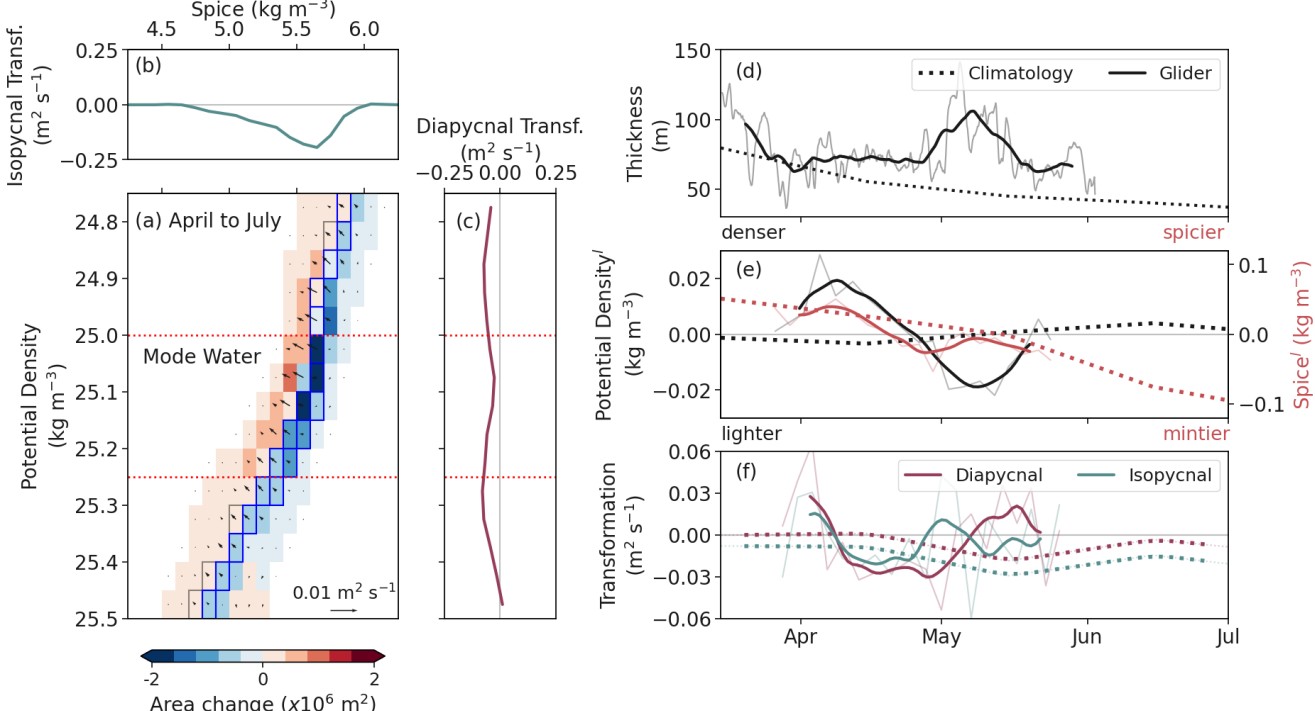

**Figure 2: Timescales of mode water transformation.** (a) Climatological area changes between April and June. Arrows depict the transformation direction and magnitude per σ-τ water class. Mode water band (σ: 25-25.25 kg m$^{-3}$) is delimited between the horizontal red dotted lines in panels (a) and (c). The σ-τ classes at the domain's boundary, thus, subjected to exchange flux, are marked with squares, colored blue when contributing more than 25% of the area loss, and red when contributing more than 25% of the area gain. (b) Integrated isopycnal transformation per spice class from panel (a). (c) Integrated diapycnal transformation per potential density class from panel (a). (d) Mode Water thickness from glider data (solid gray), after applying a 24-hour rolling mean (solid black), and the climatology (dotted). (e) Potential density anomaly (black) and spice anomaly (red) for the glider (solid light), the glider after applying a 10-day rolling mean (solid), and the climatology (dotted). The anomaly represents the anomaly from the time-mean of the period from mid-March to July. (f) Integrated diapycnal transformation (purple) and isopycnal transformation (blue) for mode water (σ: 25-25.25 kg m$^{-3}$), from glider data (solid light), the glider after applying a 10-day rolling mean (solid), and the climatology (dotted).

Applying the water-mass transformation framework to high-resolution glider sections enables quantification of transformation variability across different temporal resolutions and scales. To assess how sampling period and resolution affect the ability to reproduce the high-resolution glider mean, the 3-day transformation time series (Figure 2f) was sub-sampled over a continuous range of intervals from 3-day to monthly (Figure 3, scatters). For each interval, we generated multiple realizations of a mean by shifting the starting time iteratively by 3-day steps —for example, a 6-day sampling interval yielded two mean estimates, while a 30-day interval produced 10. The extremes of the transformations decrease after applying a low-pass filter (rolling means of different window sizes; violin plots, Figure 3). The spread in the mean



transformation (black markers) increases as resolution decreases (up to monthly climatology resolution, orange markers). This spread reflects the reduced probability of recovering the "true" high-resolution mean. If each transformation were

insensitive to the sampling frequency, its magnitude would remain close to the high-resolution mean regardless of the sampling period. From an Argo monthly climatological perspective, the probability of recovering the 3-day sampled mean within one standard error is about 60% for the diapycnal transformation and 40% for the isopycnal transformation. Using the glider sub-sampled timeseries, this likelihood drops with coarser sampling: from 75% at 6-day intervals to less than 50% at sampling frequencies longer than 12 days. These results highlight the importance of collecting data at least weekly to capture

the influence of high-frequency variability. For example, sampling only once every 30 days reduces the chance of capturing the true mean by about 85% compared to 3-day sampling.

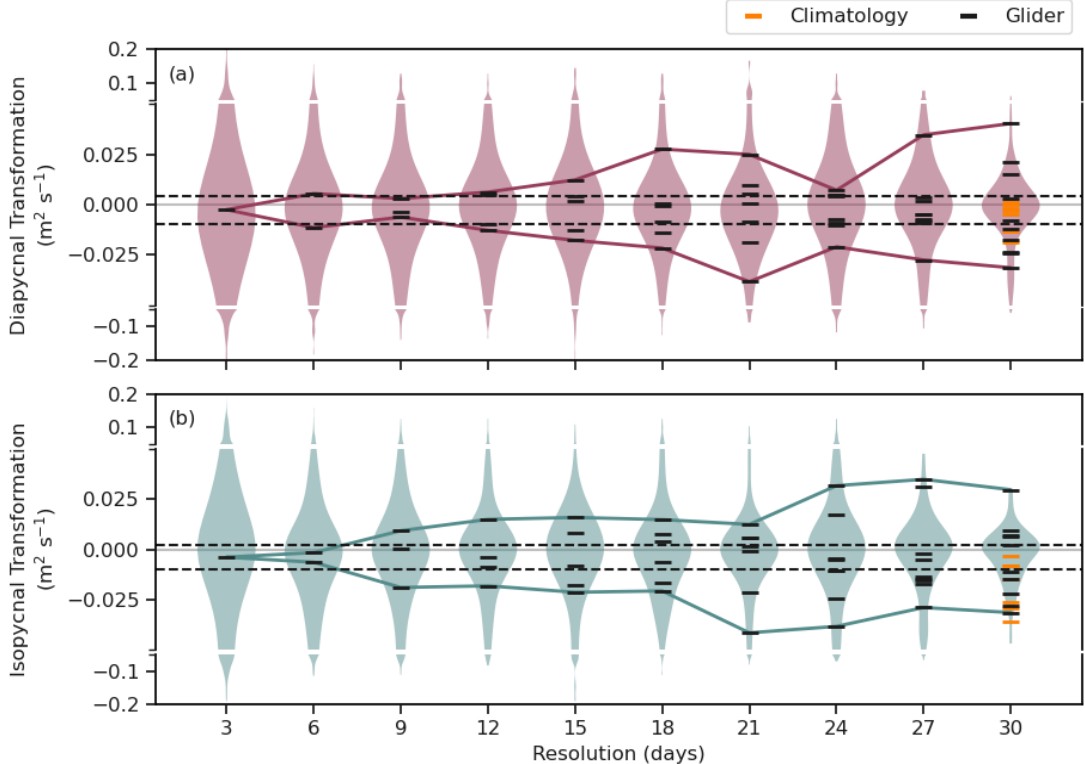

**Figure 3. Sensitivity of mean transformation estimates to sampling resolution.** (a) Diapycnal and (b) isopycnal transformations
distribution after smoothing the transformation with a resolution-day rolling mean as violin plots. Lines (-) indicate mean transformations estimated from sub-sampling the 3-day time series at coarser intervals (resolutions): black for the glider dataset, and orange for the climatological dataset. The coloured line envelopes the minimum and maximum mean transformations from sub-sampling the time series, and the black dashed lines indicate the transformations within the mean ± 1 standard error of the highest resolution (3-day).






## 3.2 Enhancement of diapycnal and isopycnal transformation by mesoscale eddies

The high-resolution observations provide an excellent dataset to analyze the temporal and spatial evolution of mesoscale
eddy activity and associated water mass transformation within the mode water from mid-March to early June. The ToEddies
Global Mesoscale Eddy Atlas (see Section 2.1, Laxenaire et al., 2024) reveals the presence and persistence of anticyclonic
and cyclonic eddies in the Sea of Oman during this period (Figure 4a), as well as the crossing of three anticyclones along the
glider's trajectory (Figure 4b). These mesoscale structures dominate the surface circulation during the sampling period and
undergo gradual displacement and structural changes over time. The glider's path through these eddies (Figure 4b) captures
the dynamic interaction between the platform and the evolving eddy field. Enhanced EKE values (Figure 4c) correspond to
periods of intensified mesoscale activity and potential eddy-glider encounters. The agreement between the satellite-derived
EKE and the glider-derived EKE confirms the presence of high-EKE periods during the campaign.

The glider-derived observations reveal the vertical imprint of these eddies on the thermohaline structure and associated water
mass transformation. The observed vertical displacements of isopycnals and intrusions of anomalous water masses (Figure
4e) are consistent with mesoscale stirring. These anomalies coincide with changes in isopycnal and diapycnal transformation
coinciding with mesoscale eddy activity (Figure 4d), suggesting an active role of mesoscale dynamics in modulating vertical
exchanges and interior ventilation. For instance, around April 6, when an anticyclone is present, a marked increase in surface
EKE (Figure 4b) corresponds with simultaneous peaks in both isopycnal and diapycnal transformation rates of change
($|\Delta Transf.|$; Figure 4c and d), suggesting intensified stirring and mixing as the glider samples through the eddy structure.

Around April 15 (Non-eddy case in Figure 5), the mesoscale eddy moves away from the glider transect, as indicated by both
the ADT maps (Figure 4a3) and the glider spice time series, where isopycnals flatten (Figure 4e). The glider trajectory
during this period does not intersect any prominent eddy (Figure 4b), suggesting that it is transiting through a relatively
quiescent background flow. Correspondingly, the transformation rates, both diapycnal and isopycnal, are stable and similar
to the ones derived from the climatology (negative and $O(10^{-2})$, Figure 2e and 4d). This calm period contrasts with the more
dynamic intervals before and after, suggesting that enhanced transformation is closely tied to eddy presence and activity. The
transformation around April 15 thus serves as an ad hoc baseline, highlighting the mesoscale-driven nature of the stronger
exchange events observed throughout the record.

A second event occurs around April 27-May 3 (Eddy case in Figure 5), when the glider crosses an anticyclone again (Figure
4a4, b, and e) and isopycnal transformation increases (Figure 4d), underscoring the dominant role of mesoscale processes in
lateral tracer transport. This is followed by a pronounced diapycnal transformation event, likely driven by vertical exchanges
at eddy boundaries and/or interaction with shelf bathymetry.




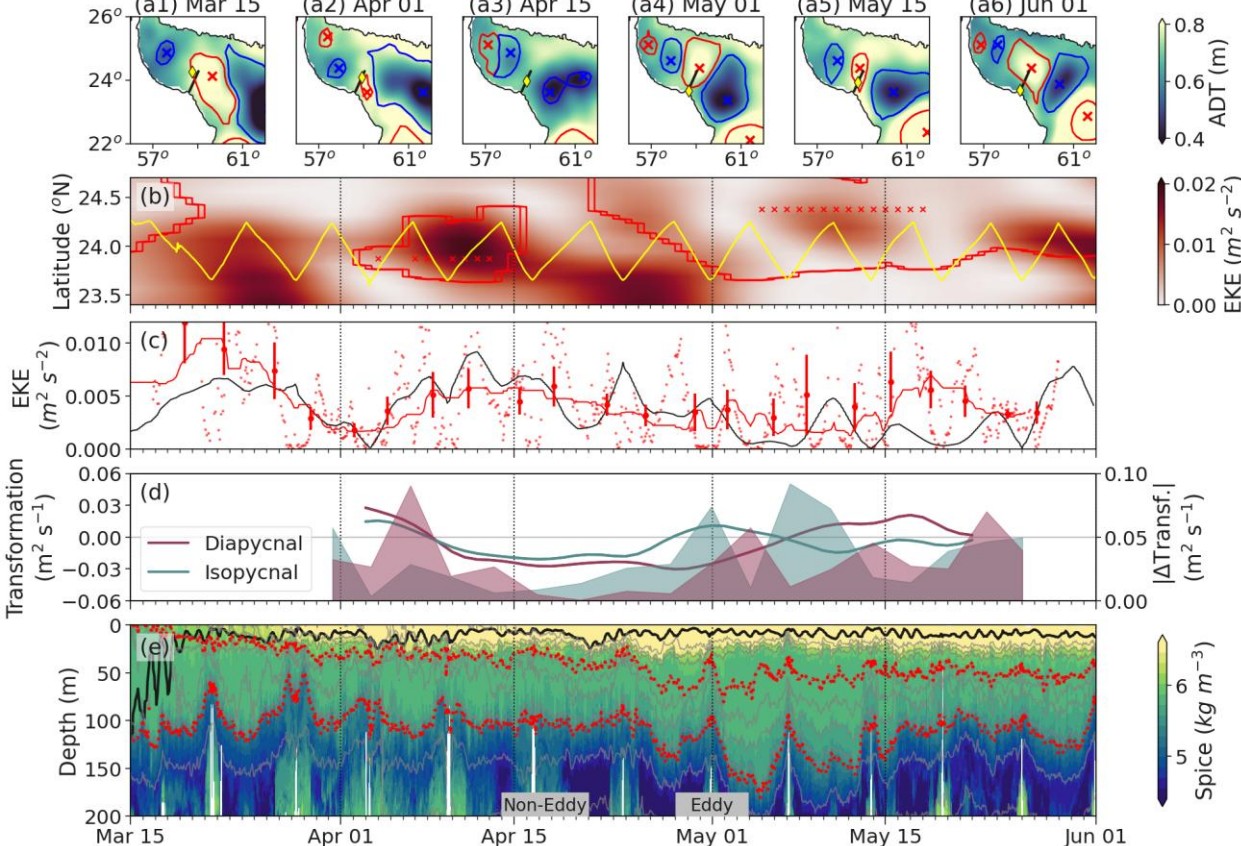

**Figure 4. Mesoscale eddies and mode water transformation.** (a1-a6) Snapshots of ADT in the Sea of Oman on the 1st and 15th day of
each month between March and June. The glider location on each date is marked with a yellow diamond, and its transect is shown as a
black line. The eddies are retrieved from TOEddies Atlas (anticyclons in red, cyclones in blue): the eddy center (x) and the maximum
extent (solid contour) are also shown. (b) Hovmoller of EKE from satellite-derived geostrophic velocities with the glider track in yellow.
The eddy center (x) and the contours of maximum extent (solid contour) are shown as in panel (a). (c) EKE retrieved from the glider dive-
average current (small red dots), after applying a 24-hour rolling mean (red line), and average per transect (large red dots) with the
standard deviation for the transect as a vertical red line. EKE retrieved from ADT (black line), scaled by a factor of 2 to match the
magnitude of the EKE from the glider. (d) Mean transformation per transect (left axis), and the absolute change of transformation over
time (|ΔTransf.|; right axis) to show when the change in transformation occurs for both diapycnal and isopycnal transformation. (e) Spice
time series with the mixed layer depth (black line), isopycnals (gray lines), and top and bottom boundaries of the core of the mode water
(25-25.25 kg m$^{-3}$; red dotted line). The two labels show the periods chosen as "Eddy" and "Non-Eddy" study cases in Figure 5.

To investigate the spatial structure of water-mass transformation and its dependence on mesoscale eddy activity, we
remapped the transformations in distance-pressure coordinates from the shelf. The transformation rates, when averaged over
the repeated sections of the entire glider deployment (Figure 5a-b) show weak diapycnal and isopycnal transformations. This
mean view closely resembles those derived from the monthly climatological dataset, despite the climatology being composed
of basin-wide Argo profiles (Figure 2a), reaffirming that glider-based observations can reproduce the climatological view
when averaged over long enough timescales, and that high-frequency variability is largely smoothed out in monthly means.



We compare the average transformation sections with two contrasting case studies: one with no eddy presence (Figure 5d-f) and one with an eddy (Figure 5g-i; The periods marked in Figure 4e). The Non-Eddy Case (Figure 5d-f) provides a baseline scenario where the glider transect occurs in a region with no identifiable eddy, as confirmed by the lack of coherent structures in the ADT maps (Figure 4a.3). Both isopycnal and diapycnal transformation rates are comparable to the

climatological mean and are relatively weak within the mode water core (~ -0.01 and 0.005 $m^2$ $s^{-1}$, respectively; Figure 2 and 5d-e).

The Eddy case (Figure 5g-i) captures a glider transect through an anticyclonic eddy. In contrast to the quiescent background conditions seen in the Non-Eddy Case, this eddy drives an order of magnitude enhancement of diapycnal transformation above and below the mode water core (~0.05 $m^2$ $s^{-1}$), pointing to intense vertical exchanges associated with eddy-induced

mixing. Isopycnal transformation is also intensified and spatially coherent along sloping density surfaces (~ 0.04 $m^2$ $s^{-1}$). The elevated isopycnal transformation rates imply vigorous lateral mixing along density surfaces, enhanced by eddy-driven stirring and frontal slumping. Such strong transformation rates suggest that the eddy acts as an efficient conduit for water-mass transformation, especially within mode water density layers. This case highlights the dynamically active role of anticyclonic eddies in modulating both vertical and lateral mixing processes, and thus promoting the redistribution of heat

and salt and contributing to the ventilation of the ocean interior.

The exchange term (grey bars in Figure 5c, f, and i), calculated as the residual in the transformation budget, serves as a proxy for advective fluxes through the section and can highlight dynamically active regions where advection dominates over mixing. In the Eddy Case, this term is particularly pronounced, indicating a strong advective component related to eddy presence (0.59 ± 0.48 $m^2$ $s^{-1}$; Figure 5i). In contrast, the exchange flux is much weaker in both the mean section and the

Non-Eddy Case (0.16 ± 0.13 $m^2$ $s^{-1}$; Figure 5c and f), suggesting a reduced role of advective processes. Eddies, both surface and subsurface, are well known to transport coherent parcels across long distances (e.g, Frenger et al. 2018), driving the strong exchange seen here. Altogether, the exchange term adds an important dimension to the volume budget, helping to quantify the net effect of non-mixing processes, and further supporting the view that mesoscale advection can be a dominant pathway for ventilation and altering stratification in the upper ocean.







**Figure 5. Case studies**. (a) Mean diapycnal and (b) isopycnal transformations in geographical space (distance from shelf vs. depth). Gray lines: isopycnals; red lines: 25.00 and 25.25 kg m$^{-3}$; dotted: iso-spice. Density classes lighter than 24.5 kg m$^{-3}$ are diagonally hatched (//) to



exclude water classes with a potential air-sea flux transformation term. (c) Diapycnal, isopycnal transformations, and exchange fluxes averaged over 0.25 kg m$^{-3}$ density bands. Panels (d-f) show a case without an eddy (Non-Eddy Case); (g-i) a case with an eddy (Eddy Case). (d, g) Diapycnal, (e, h) isopycnal transformations in geographical space; (f, i) transformations and fluxes in density space. In (d-e, g-h), dotted and solid isopycnals correspond to initial and final time steps. Vertical hatches (‖) mark regions where exchange flux accounts for > 25% of volume change. Density classes lighter than 24.5 kg m$^{-3}$ are hatched (//) to exclude water classes with a potential air-sea flux transformation term. (j-l) Spice-integrated averages of (j) diapycnal, (k) isopycnal transformations, (l) exchange flux; lines: eddy (solid), non-eddy (dashed), and Argo climatology (dotted). The averages are on the absolute values of the transformations, thus focusing on their magnitudes rather than their direction. (m) Mean density profile for eddy (solid) and non-eddy (dashed) classification, and climatology (dotted). N indicates the number of glider sections that contribute to the mean. Lines in (c, f, i, j-l) separate 0.25 kg m$^{-3}$ bands; red lines denote 25 and 25.25 kg m$^{-3}$. Note differing x-axis ranges in (c, f, i) and (j-l).

The impact of the dynamical regime on transformation magnitudes within the mode water layer is striking (Figure 5j-l). The passage of mesoscale eddies leads to a marked intensification of all three components of the volume budget within the mode water layer, reflecting enhanced isopycnal and diapycnal mixing as well as increased advective exchange (see summary schematic in Figure 6). Compared to non-eddy conditions, diapycnal, isopycnal, and exchange fluxes increase by 61%, 45%, and 66%, respectively (Figure 5j-l). In the absence of eddies, mode water undergoes significantly weaker transformation - with transformation magnitudes similar to the climatological average (Figures 1 and 5j-l). It is noteworthy that these estimates represent integrated values across the glider transect, and local transformation rates may vary substantially, enhanced or suppressed, depending on eddy position, intensity, and interactions with topography.

To further contextualize these differences, we compared the transformation rates under eddy and non-eddy conditions with the climatological mean derived from Argo profiles. Diapycnal transformation exhibits the largest relative increase during eddy periods, rising by 163% compared to the climatological mean (Figure 5j). This substantial enhancement likely reflects intensified vertical exchange driven by eddy-related processes that are absent from monthly climatologies. Even outside of eddy influence, the diapycnal transformation from the glider data remains elevated (~0.13 m$^2$ s$^{-1}$, a 63% increase relative to the climatological mean), likely due to the glider's ability to resolve finer-scale vertical structure and fluxes resulting from diapycnal mixing.

Isopycnal transformation also increases under eddy influence, with a pronounced peak in the mode water layer (~0.16 m$^2$ s$^{-1}$), corresponding to a 43% rise over the climatological mean (Figure 5k). This points to enhanced lateral stirring and strain-driven intrusions along density surfaces, characteristic of mesoscale eddy activity. In contrast, during non-eddy conditions, isopycnal transformation remains weaker and closely matches the climatological estimate (~0.11 m$^2$ s$^{-1}$). This suggests that eddy-related strain and intrusions play a major role in redistributing water masses and altering thermohaline structure within the mode water. In the absence of eddies, isopycnal transformation is reduced and comparable to the climatological mean.

The exchange component (Figure 5l) similarly shows a pronounced peak in the presence of eddies (~0.5 m$^2$ s$^{-1}$, 140% increase relative to climatology), and is notably lower under non-eddy conditions (~0.3 m$^2$ s$^{-1}$, 45% above the climatological mean). The greater exchange fluxes observed in the high-resolution glider dataset reflect the contribution of strong mesoscale advection not captured in the monthly climatologies.



## 4. Discussion

Mode waters play a crucial role as pathways connecting the surface and subsurface ocean, influencing the distribution and
transport of oxygen, heat, and remineralized organic matter. In the Arabian Sea, mode waters bound the upper oxycline of
the Arabian Sea oxygen minimum zone, thus playing a central role in shaping regional oxygen distributions. As shown by
Jutras et al. (2025), at long-term and large spatial scales, more than 50% of oxygen changes along mode water ventilation
pathways can be attributed to mixing with surrounding oxygen-poorer waters. Our results build on this by showing mode
water changes at shorter timescales, highlighting that transformation processes are strongly enhanced within the mode water
layer during periods of eddy activity.

High-resolution glider observations reveal the much greater intermittency and intensity of transformation processes missed
in climatological datasets. This challenges the underlying assumptions of climatological approaches for estimating water
mass transformation and understanding upper-ocean dynamics. Monthly climatologies, while valuable for capturing broad
seasonal trends, inherently smooth out the episodic and spatially localized processes (e.g, mesoscale eddies and
submesoscale fronts) that strongly contribute to net water-mass transformation. Our analysis shows that a significant fraction
of the total transformation, particularly within the mode water layer, occurs during short-lived eddy events, which are
effectively diluted in monthly-averaged fields (Figures 2 and 3). As a result, climatological approaches underestimate both
the intensity and variability of transformation processes, particularly the contributions from isopycnal stirring and advective
exchange.

Mesoscale eddies can trap and transport mode water, preserving its properties and prolonging the lifespan of this buffer
layer. In the Arabian Sea, eddies have been shown to modulate the distribution and characteristics of water masses such as
the Arabian Sea High Salinity Water (Thoppil, 2024; Trott et al., 2019). Moreover, anticyclonic eddies have been found to
trap more mode water than cyclonic ones (e.g., Liu et al., 2013 in the North Atlantic), contributing significantly to total mode
water transport—up to 17% south of the Kuroshio Extension (Shi et al., 2018), comparable to the transport by the mean flow
(Xu et al., 2016). We complement these findings by showing that, at short spatial and temporal scales, eddies transform
mode waters (Figure 4-6). We hypothesize that by driving intermittent but intense exchanges, eddies can influence
stratification and shape mode water transformation. While the water-mass transformation framework allows for indirect
estimation of the integrated effects of mixing, understanding the specific mechanisms behind these transformations,
particularly at fine scales, requires direct observations. Including turbulence measurements, such as microstructure-derived
diffusivities, would help disentangle the relative roles of vertical mixing, lateral stirring, and advection, and thus refine the
interpretation of transformation processes in dynamic regions like the Sea of Oman.

Mesoscale eddies play a crucial role as biogeochemical agents. Subsurface eddies can carry oxygen and nutrient signatures
over distances of hundreds of kilometers (Frenger et al., 2018) and can influence subsurface biogeochemistry over extended



periods (Karstensen et al., 2017; Schütte et al., 2016). Mesoscale eddies influence the boundaries of oxygen minimum zones
(e.g., Auger et al., 2021; Eddebar et al., 2021; Sarma et al., 2018), and it has been shown that eddy-resolving models lead to
better representation of these zones' intensity and extent (Calil, 2023; Lévy et al., 2022; Resplandy et al., 2012). The
intensified isopycnal and exchange-driven transformations we observe during eddy periods (Figure 4-6) imply stronger
horizontal and vertical transport, reinforcing the idea that they play a disproportionate role in regulating fluxes of heat, salt,
and biogeochemical tracers in the upper ocean.

We suggest that ventilation timescales for the mode water layer are highly variable in time and space. This variability has
significant implications for interpreting biogeochemical observations or budgets derived from time-averaged or coarse-
resolution data, which may underestimate the influence of short-lived but intense eddy events. Our findings raise important
questions about the timescales over which biological activity and physical mixing interact and balance across broader spatial
and temporal domains, which need to be further investigated. Capturing this level of detail is essential for improving
biogeochemical model parameterizations, which rely on an accurate representation of exchange rates and timescales.
Moreover, it provides a basis for evaluating how these processes may respond to future changes in ocean stratification, eddy
activity, and circulation patterns.



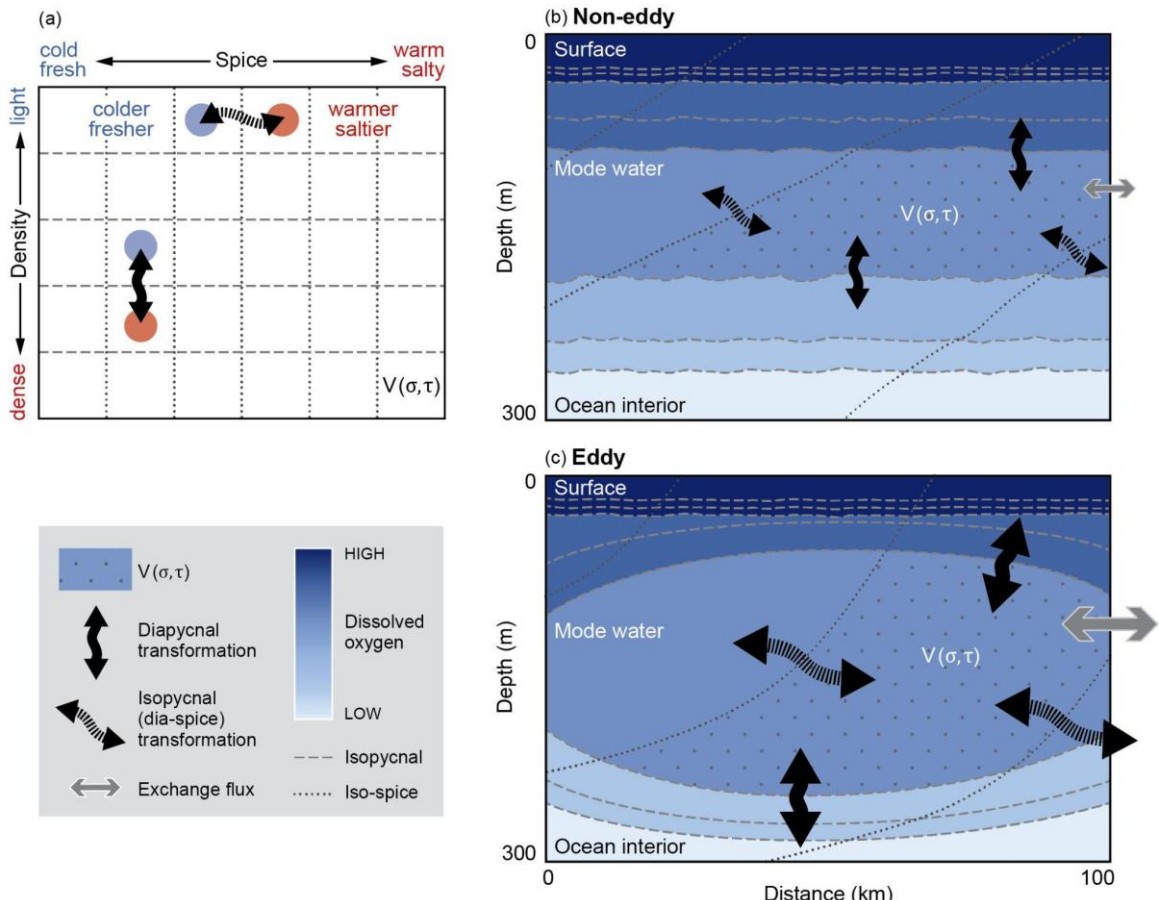

**Figure 6. Water mass transformation framework for eddy and non-eddy sections**. (a) Diagram showing the effect of the transformation of a given water class in σ-τ space (adapted from Figure 1a from Portela et al. (2020b)), and (b-c) processes involved in the volume change of a given σ-τ class in geographical coordinates for (b) a non-eddy and (c) an eddy section colored by dissolved oxygen concentration.

## 5. Conclusion

Through a detailed characterization of their temporal variability, we find that the mode water core in the Sea of Oman becomes progressively denser and mintier as its volume diminishes between March (when it is capped just below the surface) and June. By applying the water mass transformation framework in σ-τ space to both high-resolution glider data and a monthly climatological Argo dataset, we quantify the dominant timescales and transformation processes acting within the mode water layer. Isopycnal transformation rates were, on average, three times greater than diapycnal transformation. However, episodic mesoscale eddy events lead to locally enhanced diapycnal fluxes.

This framework, traditionally applied to coarse-resolution climatologies or model simulations, proves capable of capturing transformation variability and extremes that are absent from monthly climatological datasets, highlighting the unique value

of gliders. We estimate that the probability of recovering the 3-day sampled mean from monthly climatologies is between 40-60%, emphasizing how coarse temporal sampling can distort our understanding of the inherently episodic and dynamic nature of water-mass transformation. This observed variability is attributed to mesoscale eddies.

During eddy passages, all three components of the mode water volume budget -diapycnal, isopycnal transformations, and exchange flux- are significantly intensified compared to non-eddy conditions (Figure 6). Overall, our results demonstrate that transformation within the mode water layer is strongly enhanced by mesoscale activity. These findings underscore the need for sustained, high-resolution observations, as they would allow us to link integrated transformation rates more explicitly to underlying physical drivers, resolve the vertical structure of mixing processes, and assess the role of (sub-)
mesoscale dynamics and topographic interactions. Ultimately, combining transformation diagnostics with targeted in situ measurements will improve our ability to constrain ocean ventilation and tracer budgets, refine model parameterizations, and evaluate how these processes respond to a changing climate.

**Code availability.** The software associated with this paper is publicly available on GitHub (https://github.com/EstelFont/Transforamtion_Mode_Water).

**Data availability.** The Argo data used in this study is available by the International Argo Program and the national programs contributing to it for the domain 22.5-26 °N, 56-61°E and between 2000-2023 (Argo, 2023). Ocean glider data are available from the British Oceanographic Data Centre (Queste et al., 2018b). The TOEddies can be accessed at SEANOE (https://doi.org/10.17882/102877, Laxenaire et al., 2024). The bathymetry data are available from the GEBCO Compilation Group  (https://doi.org/10.5285/f98b053b-0cbc-6c23-e053-6c86abc0af7b, GEBCO, 2023). The sea surface geostrophic
velocities and absolute dynamic topography data are available from the E.U. Copernicus Marine Service Information (https://doi.org/10.48670/moi-00148, CMEMS, 2025).

**Author contribution:** EF, EP, BYQ, and SS conceptualized the study. EF performed the data curation and formal analysis and wrote the manuscript draft with input from MP. EP, BYQ, SS, and MP reviewed and edited the manuscript.

**Competing interests:** The authors declare that they have no conflict of interest.

**Acknowledgements:** We are grateful to the UEA Seaglider Facility, Sultan Qaboos University technical staff, and Five Oceans Environmental Services consultancy for their technical help with fieldwork. We thank Lionel Guez for his assistance with TOEddies detection and tracking, including help with data access and guidance on its use. We thank Gwyn Evans for the publicly available water mass transformation framework code (https://github.com/dgwynevans/wmt), from which we adapted our implementation.



**Financial Support:** EF and BYQ are supported by ONR GLOBAL Grant N62909-21-1-2008; MP and BYQ are supported by Formas Grant 2022-01536. EF and SS are supported by a Wallenberg Academy Fellowship (WAF, 2015.0186) and by the Swedish Research Council (VR, 2019-04400). S.S. has received funding from the European Union's Horizon Europe ERC Synergy Grant under grant agreement No. 101118693 (WHIRLS). EF gratefully acknowledges Adleberska Stiftelserna for partially funding a research visit to EP.

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
