# Peer review of "Spatiotemporal scales of mode water transformation in the Sea of"

_EGUsphere, 2025_

## Referee Comment (RC2)

Review: Spatio-temporal scales of mode water transformation in the

Sea of Oman

General comments

This study uses a water mass transformation framework to investigate drivers of mode water volume change in the Sea of Oman. The variables used to define water masses are potential density and spice which allows the mode water volume budget to be decomposed into isopycnal transformation, diapycnal transformation and an exchange flux across the boundary of the region considered. The methods are applied to a dataset derived from ARGO floats to produce a climatology, and data from a high resolution glider, with the aim of investigating drivers of volume changes on shorter timescales. The water mass transformation methods used in this study have not previously been applied to higher temporal resolution data making this an important study for people who may wish to carry out similar analysis in the future.

The key findings indicate that the climatology produced from ARGO floats smooth out mode water volume changes on shorter timescales. Specifically, the presence of mesoscale eddies greatly enhances isopycnal transformation, which is then followed by diapycnal transformation, over time periods shorter than a week. Such periods of high variability are not captured in the climatology produced from the ARGO data. The need for higher resolution sampling is highlighted so that shorter periods of high variability in volume changes of mode water, particularly due to the presence of mesoscale eddies, are captured. This is important both for understanding what is happening the ocean, as well as the parametrisation of such processes in models.

Overall this is a high quality and well written study that I think the community will benefit from provided the comments below are addressed with a particular focus on improving the explanation of the water mass transformation framework in section 2.2.

Josef Bisits

Specific comments

Line 15: Please clarify if the statement Mode water predominantly transforms along isopycnals is something

that is already known or a finding from this study.

1

Line 20: Is the methodological approach new? Certainly it has not been applied to higher resolution glider data but the theoretical framework was established in Evans et al. (2014).

Line 84: Please clarify if the density threshold was calculated using potential density or in-situ density.

Line 148: What is the reference pressure used to calculate potential density? If it is the same throughout the study, please include what the reference pressure is here, or with an earlier mention of potential density. Same goes for spiciness, please indicate what reference pressure is used.

Section 2.2: There are some inconsistencies in the explanation and mathematical expressions in this section which made it hard for me to follow. Equation (1), from the text, is the volume change for a specific  $\sigma - \tau$  class. On line 154 the expression  $dV/dt = A\mathbf{x}$  equates the change in volume to a linear system. On line 155 the vector  $\mathbf{x}$  is defined as  $\mathbf{x} = (U_{\sigma} + U_{\tau} + \Psi)$ , which appears to be a scalar value. This means that  $A\mathbf{x}$  is not a linear system as it is currently defined. I think what the authors mean is that each component of  $\mathbf{x}$  describes the change in volume of a specific  $\sigma - \tau$  class for a given process so  $\mathbf{x} = (U_{\sigma}, U_{\tau}, \Psi)$ . To consider all  $\sigma - \tau$  classes I think  $\mathbf{x}$  should then be  $\mathbf{x} = (U_{\sigma}^1, \dots, U_{\sigma}^m, U_{\tau}^1, \dots, U_{\tau}^n, \Psi^1, \dots, \Psi^l)$  where m and n are the number of  $\sigma$  and  $\tau$  classes, respectively, and l is the number of  $\Psi$  terms. Then, the left hand side of  $dV/dt = A\mathbf{x}$  needs to be updated, perhaps by defining the vector  $\mathbf{V}$  of all volume classes, to reflect that the result is a vector so the expression for the residuals on line 178 is consistent.

I appreciate the methods used in this study are outlined in Evans et al. (2014) and Portela et al. (2020) but this study still needs to correctly set up the theoretical framework either in the text in this section or perhaps add an appendix with the full expressions for the linear system i.e. define the matrix A and vector  $\mathbf{x}$ .

Line 186: Including the meaning of "mintier" in parentheses would be good for readers not familiar with the spice variable.

Line 234: Please clarify what the "true" high-resolution mean is.

Line 236: A probability should be between zero and one so I think replacing probability with likelihood is appropriate here (this also matches terminology to what is used on line 238).

Line 314: Would cyclonic eddies produce a similar modulation to vertical and lateral mixing? A short comment here on if there are any expected differences between cyclonic and anticyclonic eddies would be appropriate.

**Technical comments**

Line 58: I think the word "used" should be use here.

**Line 146:** Equations 1.1 and 1.2 should have the same symbol for  $\Pi$ , currently 1.1 has a bold symbol. Related

to this, could a single expression be written for  $\Pi$ ? Something like

$$\Pi(\sigma, \sigma') = \Pi(\tau, \tau') = \begin{cases}
1 & \text{if } \sigma \in \sigma' \text{ and } \tau \in \tau' \\
0 & \text{otherwise.}
\end{cases}$$
(1)

then only  $\sigma'$  and  $\tau'$  need be defined (which they are on line 147).

**Line 159:** Please update tracer surfaces to tracer *iso*-surfaces or indicate they are surfaces of constant tracer (perhaps it is implied but I think worth explicitly stating the first time).

Line: 174: I think "(two sections)" could be removed, the clarification that it is a week is sufficient.

**Line 188:** Should the second sentence be "Thus, there is *mode* water volume formed *from* ..."? As written I found it unclear.

**Line 195:** Is the word *scale* missing after smaller here?

Lines 200, 201, 204, 205: tranf. should be transf.

Line 267: This is the second use of coincide in this sentence. I think replacing "coinciding" with "along" would improve readability here.

Figure 5 caption: The description of the "red lines" could include they are the potential density range for the mode water.

Line 414: Should the first use of "their" be the?

Line 422: In line with an earlier point, I think probability should be replaced with likelihood.

**Line 426:** Expanding on what is in the parentheses would be good here, e.g. (as the schematic in figure 6 shows).

Line 429: Rather than have "(sub)" I think it worth including the word submesoscale e.g. "assess the role of submesoscale and mesoscale dynamics.."

**References**

- [1] Dafydd Gwyn Evans et al. "The imprint of S outhern O cean overturning on seasonal water mass variability in D rake P assage". In: *Journal of Geophysical Research: Oceans* 119.11 (2014), pp. 7987–8010.
- [2] Esther Portela et al. "Interior water-mass variability in the Southern Hemisphere oceans during the last decade". In: *Journal of Physical Oceanography* 50.2 (2020), pp. 361–381.

---

## Author Comment (AC1)

We thank the referee for critically reading this manuscript and providing helpful feedback, which has added a great deal to improve the manuscript and clarify the text. We respond to all issues addressed in their comments below, as well as adding the revised changes in the manuscript. The Reviewer comments are included here in black, and our answers below their respective comments in blue. The text that has been modified in the manuscript according to the reviews is presented in *italic*. The line numbers in the answers refer to the marked-up manuscript version with tracked changes

**RC1**

**General Comments**

This is a well-written and scientifically rigorous manuscript that addresses the spatiotemporal scales of mode water transformation in the Sea of Oman. The authors combine Argo climatologies with high-resolution underwater glider observations to investigate the relative roles of isopycnal and diapycnal processes in modifying mode water volume and properties. The methodological framework, based on σ-τ coordinates, is innovative and provides new insight into how mesoscale eddies influence transformation rates.

I commend the authors for the clarity of their writing, the thoroughness of their analysis, and the careful integration of climatological and glider datasets. The paper convincingly demonstrates the added value of high-resolution glider observations for capturing episodic and small-scale processes that are missed in climatologies. The conclusions about eddy-driven intensification of both diapycnal and isopycnal transformations are compelling and make a valuable contribution to our understanding of mode water variability.

Overall, this is an excellent paper that makes a valuable contribution to the field. I recommend publication after the authors consider the following specific and technical comments.

- Joe Gradone

**Specific Comments**

**Line 10:** The second sentence of the abstract is a bit of a beast to read. I only became oriented once making it to the end where you state "deeper oxygen minimum zone". I suggest moving the word "deeper" to the first mention of the oxygen minimum zone and then consider breaking this sentence up into two sentences.

Thank you for the suggestion. We applied the suggested changes and now it reads in L10 as *"This capped and well-mixed oxygenated layer decouples the oxygen minimum zone from ocean surface processes. It also provides a space for remineralisation, reducing oxygen demand in the oxygen minimum zone."*

**Line 13-14:** Initially, when I read the abstract, I was questioning how you could do this analysis on a 3-day temporal scale with a monthly climatology. The use of glider data in the analysis is clear in the main paper. I suggest maybe something like "higher resolution underwater glider observations" to distinguish the difference.

Thank you. We swapped the order of the sentence, so now it reads:

L12-15: *"We perform a volume budget analysis to investigate the mechanisms driving mode water volume change in the Sea of Oman from monthly to 3-day temporal scales using*

*monthly climatologies derived from profiling floats and high-resolution underwater glider observations."*

**Line 81:** Observations projected onto an orange line, not a blue line, correct?

Thank you. After modifying the figure, the line is black. We have modified the text (L81, 125 & 187).

**Line 82:** Is 2 km horizontally not a bit too fine for glider data?

Thank you for this comment. We agree that glider sampling does not intrinsically resolve 2-km horizontal scales, since the distance between successive profiles is typically several kilometres, depending on platform speed, dive geometry, etc. Our intention was not to imply true 2-km resolution but to choose a bin size small enough that profiles were not spatially aliased. Near the shelf, as topography changes rapidly, binned mean can skew towards shallow profiles due to the greater number of profiles.

To ensure that we did not introduce artificial high-wavenumber variability, all sections were subsequently smoothed using a 6-km horizontal running mean, which lies above the submesoscale range, while still allowing mesoscale gradients to be captured. The 2-km binning therefore, acts only as an intermediate gridding step to minimize data gaps and spatial aliasing before the physically meaningful smoothing is applied. Moreover, this choice is also coarser than the gridding strategy used in previous work in the region (e.g, Font et al. (2024) applied a 1-km grid followed by a 3-km rolling mean).

**Figure 1e:** I recognize there is a lot of information on this plot but as someone who is colorblind, I cannot fully understand what is going on. The red dotted line is difficult to see and the white and pink lines blend together.

Thank you. We have applied the changes you suggested in Figure 1. We made the MLD white solid, MW white dotted, and the oxygen contours yellow dashed. We have removed the gray density contours to simplify panel e).

[Figure]

**Line 184:** Text says Figure 1e-g, but those subplots do not exist in Figure #1

Thank you. Apologies for the mistake, those panels existed in a previous version of figure 1. Changed to Fig 1e.

**Line 186:** Define what mintier means here, not necessarily standard oceanographic knowledge. You define it well on line 201, so just consider some language to that effect here.

Thank you. We have added *"(i.e. fresher and colder)"* after mintier to explicitly state the definition (L209).

**Figure 5:** If this a pain, don't sweat it, minor comment here. It would help to further orient the reader if you could make panels c, f, and i have the 24.5 isopycnal surface approximately in line with the depth where the hatching stops in the corresponding plots to the left.

Thank you for the suggestion. We have tried your suggestion, but it squeezes the rest of the density bands too much, reducing their visibility, so we have decided to keep it as it was.

**Figure 5 cont.:** I cannot tell the difference between the colored bars for "diapycnal", "isopycnal", and "exchange" in panels c, f, and i. **Figure 5 cont.:** Helpful for the reader if you could put either a title or a small line of text, sort of like a legend, showing j-l correspond to diapycnal, isopycnal, and exchange, though the change in colors will also likely help this a ton.

Thank you. We have applied the changes you suggested in Figure 5 (now figure 6) and changed the colors consistently across the other figures to be colorblind friendly. We apologize for not having taken this into account. We have also added a small title in each of the panels j-l.

[Figure]

**Lines 316**: Since the exchange term is computed as a residual, uncertainties in isopycnal and diapycnal terms will propagate directly into this estimate. Could you expand on how robust this separation between mixing and advective exchange is, and whether the relative magnitudes may be sensitive to error?

We agree that diagnosing the exchange term $\Psi$ as a residual would imply that uncertainties in the isopycnal and diapycnal transformation terms will propagate into this estimate. In our implementation, however, $\Psi$ is not computed after the fact from a difference, but is solved simultaneously with $U_\sigma$ and $U_T$ in the linear inverse system $dV/dt = Ax$, where $x = (U_\sigma, U_T, \Psi)$. This means that all three components are constrained jointly by the $\sigma$-$T$ volume tendencies and by the spatial coherence imposed by the least-squares solution, rather than $\Psi$ acting as a simple residual/noise .

Two facts give us confidence that the separation between mixing and exchange is reasonably robust:

1) Small residuals of the inverse problem: Order of $10^{-5}$ for both the glider and climatological datasets, indicating that Ψ is not dominated by numerical noise or large unresolved imbalances; and

2) Consistent spatial and dynamical structure: The largest Ψ values occur where we independently expect strong advection: in the eddy case, the exchange term peaks within the mode-water density range and co-locates with strong ADT anomalies, enhanced EKE, and intensified isopycnal/diapycnal transformations. In contrast, during non-eddy conditions, Ψ is much weaker and comparable to the climatological mean. The eddy vs. non-eddy differences in Ψ are larger than the typical spread associated with our inversion residuals, suggesting that the relative magnitudes we report are robust even if the absolute values of Ψ carry some uncertainty.

We now explicitly state in the detailed Supplementary Information that Ψ should be interpreted as an *effective* advective exchange term that may also include any small residual imbalance between mixing and volume tendency: *"To note, because the exchange term Ψ is obtained as part of the least-squares solution to [5] rather than by a simple difference of diagnosed terms, it is constrained jointly with Uσ and Uτ. We therefore interpret Ψ as an effective advective exchange term."*

**Line 365:** Consider rewording/expanding to include a more general term, such as tracers. Maybe adopt the wording from line 398-399.

Thank you for this suggestion. We agree that the sentence can be broadened to reflect that eddy-driven transformations influence not only oxygen but also a wider suite of tracers. We have reworded the sentence to adopt terminology consistent with Lines 398-399 (now L426) and to generalize the statement beyond oxygen alone.

L391: *"'Mode waters play a crucial role as pathways connecting the surface and subsurface ocean, influencing the distribution and transport of tracers, including heat, salt, oxygen, and other biogeochemical properties."*

**Technical Comments**

**Line 84:** I would expect a 6 km running mean to filter out submesoscale variability. With the Rossby radius of deformation at this latitude (O) 20 km, can you comment on whether this reduces your ability to resolve the lower end of the mesoscale as well?

Thank you for raising this point. Our intention with the 6 km running mean was specifically to suppress submesoscale variability and small-scale noise, rather than to fully resolve the smallest mesoscale features.

A 6 km window strongly dampens structures with horizontal scales ≲ O(5-10 km), i.e. submesoscale variability that we do not aim to interpret with the present framework. Mesoscale features with scales comparable to or larger than the Rossby radius (≳ 20 km) remain resolved by several independent grid points after smoothing. In other words, the lower end of the mesoscale band is somewhat smoothed but not removed, and the eddy signatures we analyze (radius and deformation scales well above 20 km) are still clearly captured in the glider sections and in the derived transformation fields. Our conclusions regarding enhanced transformation during eddy passages are therefore based on features that are comfortably larger than the effective smoothing scale.

We have added a brief clarification in the Methods to state that the 6 km running mean is chosen as a compromise: it removes submesoscale variability and sampling noise while retaining the mesoscale structures that are the focus of this study.

*L84: "The 6 km running mean is selected as a compromise between filtering out submesoscale variability ($\lesssim O(10\ km)$) and preserving mesoscale structures ($\gtrsim O(20\ km)$, comparable to the local Rossby radius)."*

**Line 87:** An equation would be nice for both EKE calculations.

Thank you for the suggestion. We have expanded on the definition. Now it reads as:

*L89-94: "Eddy kinetic energy (EKE) was estimated from two independent sources: (1) the depth-averaged currents (DAC) derived from the glider flight model as $EKE = (DAC - \underline{DAC})^2$, where $\underline{DAC}$ is the mean DAC during the glider campaign (Frajka-Williams et al., 2011); and (2) from sea surface height-derived surface geostrophic velocities from satellite observations. EKE is defined as $u'^2 + v'^2$, where $u' = u - \underline{u}$ and $v' = v - \underline{v}$ are respectively the zonal and meridional velocity anomalies, where u and v are the surface geostrophic velocities on the glider transect, and $\underline{u}\ and\ \underline{v}$ are the mean surface geostrophic velocities during the glider-sampling period. "*

**Line 88:** Can you elaborate/clarify on what you mean re: "anomalies of the dive-averaged currents derived from the glider flight model"? Anomaly relative to what?

We have removed "the anomaly of…" as adds confusion and now is explicitly stated in form of equation how we derive EKE from DAC. We have modified the text in L94 to "*(1) the depth-averaged currents (DAC)...*".

**Line 100:** A 200 km buffer from the across-Gulf transect for remapping seems very wide.

We agree that 200 km may appear large relative to the width of the transect itself. Our rationale is that the goal of the Argo-based climatology is *not* to reproduce finescale cross-shelf structure, but to obtain a robust monthly mean thermohaline structure representative of the broader Sea of Oman. Argo coverage in this region is sparse and unevenly distributed (as shown in Figure 1a), and a narrower buffer (e.g., 50-100 km) results in strong spatial aliasing and insufficient profile density for several months. The 200 km radius ensures adequate sampling density across all months of the climatology while still being small enough to avoid drawing profiles from outside the Sea of Oman basin. Thus, this radius is chosen to maximize sampling coverage while retaining basin-level representativeness rather than local fidelity

To note: 1) The profiles are first remapped onto the across-Gulf transect using median binning, which strongly limits the influence of outliers or spatial inhomogeneities. 2) The climatology is then binned horizontally at 3 km, which removes residual small-scale variability and ensures that basin-scale gradients (rather than local anomalies from individual Argo floats) dominate the mapped fields. 3) Our intention is not to interpret cross-shelf signals from the climatology, but to compare large-scale, monthly transformation tendencies with the high-resolution glider data.

We have modified the description of how we produce the climatology (in response to Editor Comment) and added a clarification in the Methods to explain more explicitly why this threshold of 200km is used in response to your comment.

*L102: "…Argo profiles within a 200 km distance from the across-Sea of Oman transect (Figure 1a, orange dashed line) were selected. This strategy ensures sufficient monthly sampling coverage in this sparsely observed region to construct an across-gulf monthly climatology…"*

**Line 122:** The spice coordinate captures the isopycnal change more so than using potential density as a coordinate, no? Consider rewording this sentence.

Thank you, we agree. We have removed the "isopycnal nature of mechanisms" so now it only refers to the diapycnal mechanisms (L132).

**Line 190:** I am a little confused at how the thinning of mode waters results in a densification, but the signs of both the isopycnal and diapycnal transformation are negative (Line 192-193), implying a reduction in density.

We recognize that the relationship between thinning, mintification, and densification may not have been sufficiently clear. The key point is that the instantaneous signs of the isopycnal and diapycnal transformations (negative values) refer to net mixing tendencies at the boundaries of the σ-τ classes, not the depth-mean changes of σ and τ within the mode-water layer. In the climatological analysis, the integrated tendency over the full April-June period produces: net mintification (Δτ < 0), and net densification (Δσ > 0), as shown in Fig. 2e (now 3e). By contrast, the negative isopycnal and diapycnal transformation values in Fig. 2b-c (now 3a,c) indicate loss of volume from the lighter/spicier edges of the mode-water layer, not that the whole layer is becoming lighter. Because mode water is simultaneously thinning, this preferential loss of lighter/spicier classes results in a net shift of the remaining volume toward slightly denser classes, even though the instantaneous transformations point toward lighter isopycnals.

In other words: Transformation signs describe the direction of water-mass fluxes into/out of σ-τ bins, whereas Δσ and Δτ describe the trajectory of the remaining volume-weighted mean properties. We have reorganized that paragraph and clarified this in the text to avoid confusion.

*L211: "...On average, the isopycnal transformation dominates (-0.015 ± 0.009 m2 s-1), with a magnitude approximately three times larger than the diapycnal transformation (-0.005 ± 0.008 m2 s-1) (Figure 3a, c, and f). The signs of the transformation terms reflect the direction of the fluxes between density and spices classes, whereas the net Δσ and Δτ reflect the evolution of the volume-weighted mean properties…. "*

**Timescale of transformation of mode water section:** I found the inclusion of both the climatological analysis and the higher resolution glider data on the same plots in Figure 2 to be a lot to unpack. Similarly, while I think the title of this section is a nice description, the first paragraph could use some additional language to highlight the time period it refers to. Similar to how the second paragraph highlights how the glider data allows for a higher resolution analysis. I don't necessarily think the two paragraphs warrant their own section, but the differences in the findings are noteworthy enough to warrant additional descriptive text, at a minimum. Initially, I was going to suggest breaking Figure 2 up into two different figures, but I do find the comparison to the climatology to be helpful. The additional text in the results section will likely make the figure more digestible.

Thank you for this thoughtful comment. We agree that Figure 2 (now Figure 3) contains a large amount of information, and that the contrast between climatological and high-resolution glider transformations merits clearer framing in the text. We chose to keep both datasets in a

single figure because the side-by-side comparison is central to demonstrating how temporal averaging shapes the interpretation of mode-water transformation. However, we have revised the text to better guide the reader through the distinct timescales represented and to clarify that the first paragraph refers specifically to the *monthly climatological* perspective, while the second paragraph addresses the *intraseasonal* variability resolved by the glider.

To improve clarity, we have added explicit transitions outlining: (1) the temporal window and resolution addressed in each paragraph, (2) why the climatological and glider analyses differ. This additional contextual text helps make Figure 2 more digestible without requiring a split into two separate figures.

*L202: "We examine the seasonal-scale evolution of mode-water properties and transformations using the monthly Argo climatology (April–June). This climatological view captures the broad, low-frequency changes as the mode water evolves through late spring and early summer, but necessarily smooths over shorter-term variability..";*

*L221: "In contrast to the climatological perspective, the high-resolution glider time series resolves submonthly variability and therefore reveals the episodic, intraseasonal changes in mode water structure and transformations that are absent from the monthly climatology. This allows us to directly quantify short-lived events associated with transient processes, such as mesoscale eddies."*

**Line 368:** While I understand a large aspect of the importance of Arabian Sea mode waters is their influence on subsurface oxygen concentration, I find the discussion around your results in the context of prior oxygen-focused literature to be too direct, as it does not actually utilize any oxygen data in your analysis. Simply, the last sentence of the first paragraph in the Discussion section can either be reworded or expanded to better reflect which aspect of Jutras et al. (2025)'s study your results expand. Then, more explicitly, how one might infer the resulting changes/implications in oxygen concentration from your findings. It is clear how your findings are focused on shorter timescale changes in mode waters, but I find this important paragraph in need of larger clarification.

We agree that our original wording placed too strong an emphasis on oxygen given that our analysis does not explicitly use oxygen observations. Our intention was to situate the physical transformation processes we diagnose within the broader biogeochemical context established by Jutras et al. (2025), who quantified how mixing along mode-water ventilation pathways shapes long-term oxygen changes. We have therefore reworded and expanded this part of the Discussion to (i) clarify exactly how our results complement Jutras et al. (2025), and (ii) more explicitly outline how changes in physical transformation at short timescales could affect oxygen without overstating what we quantify directly.

*L392: "For instance, in the Arabian Sea, mode waters bound the upper oxycline of the Arabian Sea oxygen minimum zone, thus playing a central role in shaping regional oxygen distributions (Font et al., 2025). As shown by Jutras et al. (2025), at long-term and large spatial scales, more than 50% of oxygen changes along mode water ventilation pathways can be attributed to mixing with surrounding oxygen-poorer waters. While we do not diagnose oxygen directly, our results extend this understanding by showing that the physical drivers of such mixing are strongly intensified at short timescales during eddy activity. These episodic but vigorous transformation events likely modulate ventilation efficiency and tracer redistribution within mode waters."*

---

## Author Comment (AC2)

We thank the referee for critically reading this manuscript and providing helpful feedback, which has added a great deal to improve the manuscript and clarify the text. We respond to all issues addressed in their comments below, as well as adding the revised changes in the manuscript. The Reviewer comments are included here in black, and our answers below their respective comments in blue. The text that has been modified in the manuscript according to the reviews is presented in *italic*. The line numbers in the answers refer to the marked-up manuscript version with tracked changes

**RC2**

**General comments**

This study uses a water mass transformation framework to investigate drivers of mode water volume change in the Sea of Oman. The variables used to define water masses are potential density and spice which allows the mode water volume budget to be decomposed into isopycnal transformation, diapycnal transformation and an exchange flux across the boundary of the region considered. The methods are applied to a dataset derived from ARGO floats to produce a climatology, and data from a high resolution glider, with the aim of investigating drivers of volume changes on shorter timescales. The water mass transformation methods used in this study have not previously been applied to higher temporal resolution data making this an important study for people who may wish to carry out similar analysis in the future.

The key findings indicate that the climatology produced from ARGO floats smooth out mode water volume changes on shorter timescales. Specifically, the presence of mesoscale eddies greatly enhances isopycnal transformation, which is then followed by diapycnal transformation, over time periods shorter than a week. Such periods of high variability are not captured in the climatology produced from the ARGO data. The need for higher resolution sampling is highlighted so that shorter periods of high variability in volume changes of mode water, particularly due to the presence of mesoscale eddies, are captured. This is important both for understanding what is happening the ocean, as well as the parametrisation of such processes in models.

Overall this is a high quality and well written study that I think the community will benefit from provided the comments below are addressed with a particular focus on improving the explanation of the water mass transformation framework in section 2.2.

Josef Bisits

**Specific comments**

**Line 15:** Please clarify if the statement Mode water predominantly transforms along isopycnals is something that is already known or a finding from this study.

It is a finding from our study. We have changed the sentence to explicitly state that these are our results. Now it reads as follows in L15: *"Our study shows that mode water..."*

**Line 20:** Is the methodological approach new? Certainly it has not been applied to higher resolution glider data but the theoretical framework was established in Evans et al. (2014).

Thank you. We agree and have rephrased the sentence to state it*:*

*L19: " This study provides a novel application of the water mass transformation framework to high-resolution underwater gliders, and shows that this methodology can be used at higher resolution than traditional climatological products or models."*

**Line 84:** Please clarify if the density threshold was calculated using potential density or in-situ density.

We used potential density. We have clarified it now as: *"...potential density threshold of..."* (L86).

**Line 148:** What is the reference pressure used to calculate potential density? If it is the same throughout the study, please include what the reference pressure is here, or with an earlier mention of potential density. Same goes for spiciness, please indicate what reference pressure is used.

Thank you. We have clarified in the methods: *"Potential density and spice are referenced at 0 dbar."* (L88).

**Section 2.2:** There are some inconsistencies in the explanation and mathematical expressions in this section which made it hard for me to follow. Equation (1), from the text, is the volume change for a specific σ - τ class. On line 154 the expression dV/dt = Ax equates the change in volume to a linear system. On line 155 the vector x is defined as x = (Uσ + Uτ + Ψ), which appears to be a scalar value. This means that Ax is not a linear system as it is currently defined. I think what the authors mean is that each component of x describes the change in volume of a specific σ − τ class for a given process so x = (Uσ,Uτ , Ψ). To

consider all σ - τ classes I think x should then be $\mathbf{x} = \left( U_\sigma^1, \ldots, U_\sigma^m, U_\tau^1, \ldots, U_\tau^n, \Psi^1, \ldots, \Psi^l \right)$

where m and n are the number of σ and τ classes, respectively, and l is the number of Ψ terms. Then, the left hand side of dV/dt = Ax needs to be updated, perhaps by defining the vector V of all volume classes, to reflect that the result is a vector so the expression for the residuals on line 178 is consistent.

I appreciate the methods used in this study are outlined in Evans et al. (2014) and Portela et al. (2020) but this study still needs to correctly set up the theoretical framework either in the text in this section or perhaps add an appendix with the full expressions for the linear system i.e. define the matrix A and vector x.

Thank you. We apologise for the inconsitencies in the mathematical expressions and explanation of the method. We have decided to add a detailed explanation in the Supplementary Information that includes the equations and provides a more clear outline. Please see the new version Supplementary Information.

To address your suggestion, we have changed the explanation to not only one sigma-spice class, but written the expression for all sigma-space classes in a vector form as per Evans et al., 2014. We have made the changes in the manuscript accordingly (See Section 2.2, L165):

*"Using equation (1), a set of linear equations can be built to link the volume trend to the interior water-mass transformation in σ-τ coordinates as dV/dt=Ax, where dV/dt is the observed change in volume of each sigma-spice class, divided by the relevant time interval; A is the matrix of coefficients of the linear equations; and x is the vector of the resulting diasurface transformations and exchange flux. This system is solved by means of a least squares regression for the unknown transformation and exchange flow. The detailed methodology has been added to the Supplementary Information. The solution x was then*

*decomposed into the transformation across spice and density surfaces and the exchange flux across the geographical domain."*

**Line 186:** Including the meaning of "mintier" in parentheses would be good for readers not familiar with the spice variable.

Thank you. We have clarified in L209*: "mintier (i.e., fresher and colder)".*

**Line 234:** Please clarify what the "true" high-resolution mean is.

We have changed the "true high-resolution mean" to *"the 3-day mean"* in lines 267 and 273.

**Line 236:** A probability should be between zero and one so I think replacing probability with likelihood is appropriate here (this also matches terminology to what is used on line 238).

Thank you. We have changed it. (L267)

**Line 314:** Would cyclonic eddies produce a similar modulation to vertical and lateral mixing? A short comment here on if there are any expected differences between cyclonic and anticyclonic eddies would be appropriate.

Thank you for raising this point. The aim of this work is to highlight the general role of mesoscale eddies in modulating vertical and lateral mixing. Our current datasets (and scope of the study) cannot be used to explicitly distinguish between the effects of cyclonic and anticyclonic rotation of eddies on the processes we describe. Most notably, our glider dataset intersects anticyclonic eddies only, thereby our interpretations are constrained to this polarity. Given ongoing data collections efforts, it may be possible in the future to address this topic and complete a further investigation.

**Technical comments**

**Line 58:** I think the word "used" should be use here.

Changed, thank you.

**Line 146:** Equations 1.1 and 1.2 should have the same symbol for Π, currently 1.1 has a bold symbol. Related to this, could a single expression be written for Π? Something like

$$\Pi(\sigma, \sigma') = \Pi(\tau, \tau') = \begin{cases} 1 \text{ if } \sigma \in \sigma' \text{ and } \tau \in \tau' \\ 0 \text{ otherwise.} \end{cases}$$

then only σ′ and τ′ need be defined (which they are on line 147).

Thank you. We have changed Π to "not bold" for both equations and also written Π as a single expression as you suggested as Eq. 2. (L158)

**Line 159:** Please update tracer surfaces to tracer iso-surfaces or indicate they are surfaces of constant tracer (perhaps it is implied but I think worth explicitly stating the first time).

Thank you. Updated to tracer iso-surfaces (L173)

**Line: 174:** I think "(two sections)" could be removed, the clarification that it is a week is sufficient.

Removed.

**Line 188:** Should the second sentence be "Thus, there is mode water volume formed from ..."? As written I found it unclear.

Thank you. We have changed it to your suggestion (L214).

**Line 195:** Is the word scale missing after smaller here?

Yes, we have added it (L220).

**Lines 200, 201, 204, 205:** tranf. should be transf.

We have changed all of them accordingly.

**Line 267:** This is the second use of coincide in this sentence. I think replacing "coinciding" with "along" would improve readability here.

We have changed it to "along".

**Figure 5 caption:** The description of the "red lines" could include they are the potential density range for the mode water.

Changed to L363: *"red lines denote the potential density range of mode water (25 and 25.25 kg m$^{-3}$)."*

**Line 414:** Should the first use of "their" be the?

Changed.

**Line 422:** In line with an earlier point, I think probability should be replaced with likelihood.

Thank you. Changed (L445).

**Line 426:** Expanding on what is in the parentheses would be good here, e.g. (as the schematic in figure 6 shows).

We have changed it as you suggested (L449).

**Line 429:** Rather than have "(sub)" I think it worth including the word submesoscale e.g. "assess the role of submesoscale and mesoscale dynamics.."

Explicitly included as you suggested (L453).

**References**

[1] Dafydd Gwyn Evans et al. "The imprint of Southern Ocean overturning on seasonal water mass variability in Drake Passage". In: Journal of Geophysical Research: Oceans 119.11 (2014), pp. 7987-8010.

[2] Esther Portela et al. "Interior water-mass variability in the Southern Hemisphere oceans during the last decade". In: Journal of Physical Oceanography 50.2 (2020), pp. 361-381.

---

## Author Comment (AC3)

We thank the referee for critically reading this manuscript and providing helpful feedback, which has added a great deal to improve the manuscript and clarify the text. We respond to all issues addressed in their comments below, as well as adding the revised changes in the manuscript. The Reviewer comments are included here in black, and our answers below their respective comments in blue. The text that has been modified in the manuscript according to the reviews is presented in *italic*. The line numbers in the answers refer to the marked-up manuscript version with tracked changes

**RC3**

This study employs glider transect observations from the Sea of Oman to quantify diapycnal and isopycnal mixing within the mode water layer across multiple timescales. The authors compare these estimates with monthly climatologies derived from Argo float data, demonstrating the limitations of coarser temporal resolution in capturing mixing variability. Their analysis underscores the need for additional glider-based field campaigns and direct turbulence measurements. The higher-resolution glider observations provide a more complete characterization of mixing processes under both eddy and non-eddy conditions, revealing that 40-60% of transformation variability is obscured by climatological averaging.

The novelty of this work lies in its use of high-frequency observational data to estimate mode water transformation rates, offering an observational perspective that complements traditional climatological approaches. I recommend this manuscript for publication. It presents a rigorous and well-articulated analysis that advances understanding of mode water transformation processes and remains accessible to readers less familiar with water mass transformation frameworks.

**L44**: What is the volume, what is the residence time? Provide relevant values for quick mental reference for readers.

Thank you. We provide a range of MW thickness and residence time from Font et al., 2025:

L43: *"...(thickness > 50 m and residence time > 4 months; Font et al., 2025),..."*

**L48-50:** What does respiration within the water mass mean? Can you be more explicit; for ex., does respiration in this context mean physical transport of the MW? If so, how does transport within the WM lead to oxygen changes if the WM is defined to be a homogenous parcel?

We apologise for the confusion. We meant "*net community respiration*", i.e., biological consumption, not physical transport). We have clarified in the text L48.

**L53-55:** If you can't provide explicit values for volume and residence time, then in that first sentence maybe quickly mention the poorly constrained nature of that info.

Thank you. Font et al., 2025 provide explicit estimates of mode-water volume and residence time. Yet, the interannual and sub-monthly variability of these quantities due to limited high-resolution observations remains unconstrained. We have rephrased to be explicit:

L51: *"Understanding the physical processes that mix and stratify mode water across scales, and how these processes interact with biogeochemical dynamics, is critical. While Font et al. (2025) provide valuable estimates of mode-water volume and residence time, the subseasonal and small-scale variability that governs changes in its properties remains poorly constrained, largely because observations at the necessary spatiotemporal resolution are still scarce."*

**L60**: Can you elaborate on why you chose potential density as your coordinate? I don't know what "natural" means here. You explain spice well in L63, please apply this level of explanation for why you chose sigma as your other coordinate - why not neutral density?

Thank you for pointing this out. We agree that the term "natural" was not sufficiently clear.

Choosing a density-spiciness framework we are able to address isopycnal and diapycnal transformations, and to identify changes of water masses spreading along isopycnals which would be otherwise hidden. Potential density is used because it is materially conserved under adiabatic and isohaline motions and therefore provides a practical approximation to isopycnal surfaces in the upper ocean. Neutral density is more exact in theory, but is not well defined near the surface and requires full 3-D fields; even though 2D approximations can be made. On the spatial and density scales of this study (upper 200 m, $O(0.1$ kg m$^{-3})$ density range), thermobaricity (the pressure-dependence of the thermal expansion coefficient that causes potential-density surfaces to diverge from true neutral surfaces) is weak, so potential and neutral density differ only minimally and would not alter the transformation results.

We have now clarified in the text L60: "*Potential density is used because it is materially conserved under adiabatic and isohaline motions and therefore provides a practical approximation to separate isopycnal and diapycnal fluxes, while spice is...*"

**L69:** Strong motivation for this paper!

Thank you.

**L73:** Great list of questions, organized!

Thank you again!

**L82**: typo: "at"

Changed. Thank you.

**Fig 1a:** Grey dashed line that float data are projected onto...does that mean farthest points on the map are also projected onto that line? The farther ends have little to no float data, and some cross shelf from 100-1000m, is it representative of the space (in x,y and z) MW would expect to occupy?

Thank you for raising this point. The orange dashed line is used solely as the reference transect onto which Argo profiles are horizontally projected to build the across-Gulf climatology. We acknowledge that some offshore points lie far from the line, but two factors ensure that the projected climatology remains representative of the water masses sampled by the glider (See response to RC1-L100, and Editor Comment).

We restrict Argo profiles to those deeper than 1000 m and within 200 km of the transect, which eliminates shallow shelf profiles and limits contributions to deep offshore profiles that are representative of the open Sea of Oman. The resulting climatology represents a large-scale, basin-averaged cross-section rather than a precise reproduction of the glider geometry, which is exactly what we require for comparison to seasonal-scale Argo-based transformations.

We have added a short clarification in the manuscript regarding your point in the RC1 and Editor Comment. :

L101: *"...Argo float coverage spans the entire domain, with an average density of 28 profiles per 0.25° × 0.25° grid cell (Figure 1a) and a relatively uniform monthly distribution (Figure*

*1c). Argo profiles within a 200 km distance from the across-Sea of Oman transect (Figure 1a, orange dashed line) were selected. This strategy ensures sufficient monthly sampling coverage in this sparsely observed region to construct an across-gulf monthly climatology. Moreover, to avoid the influence of shallow profiles on the continental shelf, profiles shallower than 1000 m were excluded, so the transect remains representative of the environment where mode water forms and persists. Each profile was then orthogonally projected onto the nearest point along the transect (the orange dashed line in Figure 1a), which provides its along-transect coordinate. Profiles were vertically interpolated onto a uniform 2-m pressure grid, and all projected profiles were median-binned into 3-km horizontal bins along the transect. Averaging was performed on pressure levels. Monthly climatologies were produced by taking the median across all profiles within each (depth, distance) bin for each month between 2000 and 2023. This gridded product is then used as input for the σ-τ water-mass transformation calculations."*

**Fig 1d:** stability of upper 100m highly variant esp during the transition from spring to summer. Flips sign sometime in Feb/Mar. What determines the shape of that seasonal spiral (when tracking the vertices of the thermohaline stability)?

Thank you for this interesting question. The spiral-like shape in Fig. 1d arises directly from the seasonal evolution of upper-ocean stratification (N²) during winter mixing and subsequent restratification. The key drivers are winter convective mixing (Jan-Feb) (strong surface cooling and wind-driven mixing homogenize the upper layer), spring restratification (late Feb-March) (as surface heating begins), onset of strong thermal stratification (March-June; Surface warming dominates), and capping of mode waters beneath this strong surface stratification. This has been described in Font et al. 2022 and Font et al. 2025.

**Fig1e:** Grey lines are very hard to discern, please consider a different color that will pop out from the noisy background (cyan?). Can you tell from this view of the chances the MW will be subducted or mixed back up into the surface? Perhaps the time of year indicates the likelihood skewed towards mixing with deeper water masses?

Thank you. We have removed the density contours in grey to simplify the figure and have applied the changes in Figure 1 that RC1 suggested (See RC1 and Fig1). This panel primarily illustrates vertical displacements and eddy-driven modulation of the capped layer, rather than active subduction or re-entrainment. However, it is possible to infer that when the mode water layer is well beneath the MLD, and that the stratification increases, the likelihood of being re-entrained into the mixed layer is low. This occurs progressively from the end of March (capping). By the end of April the seasonal thermocline is already established, the stratification has substantially increased, and the observed variability is dominated by mesoscale-induced variability.

**L122:** Can you say the information succinctly instead of saying the rest of this sentence?

We have modified the statement (L131-134) in response to RC1-L122 and EC-L85. Further details provided in the respective responses to RC1 and EC.

**L129**: why formation and not transformation? it is the sum of formation and destruction (i.e. transformation). Later in L131 you say it's the convergence/divergence represented by the sum, so to also consider destruction/divergence it is more apt to say transformation.

Thank you for the clarification. We agree that "transformation" is the more accurate term here, since $\sum U(\sigma,\tau)$ represents the net effect of both formation and destruction (i.e., convergence and divergence) within a σ-τ class. We have replaced "formation" with

"transformation" to maintain consistency with our definitions and with the wording used later in the section (L138).

**L134**: What about the southeast part?

Thank you for raising this point. In our formulation, we allow exchange only through the northern boundary of the transect. The southern end lies close to the continental shelf, where flow is topographically constrained and cross-shelf exchange is expected to be strongly limited. For this reason, and because the northern boundary sits in the open interior of the Sea of Oman, we assume that the majority of through-section exchange occurs there. We now state this explicitly as a caveat in the manuscript.

L144: *"The southern end is shelf-constrained, cross-section exchange there is assumed to be negligible. "*

**L138**: good summary statement

Thank you.

**Eq1.1:** (looking for clarification here) Sum of (sigma bins x cumulative product of spice bins x sigma velocity)?

Thank you for pointing this out. We realize the notation in Eq. 1.1 (now 2.1-2.2) may give the impression of a cumulative product of σ and τ bins, which is not the case. The expression defines the flux across a σ-surface, integrated over the area of all grid cells that fall within a given σ-τ class. We had a mistake in the notation (the equality was in the denominator of the integral), which we have fixed for eq. 2.1 and 2.2, which now read as:

$$U_\sigma(\sigma,\tau) = \int_{\sigma'=\sigma} \Pi(\tau,\tau') \cdot u_\sigma \cdot dA \text{ and } U_\tau(\sigma,\tau) = \int_{\tau'=\tau} \Pi(\sigma,\sigma') \cdot u_\tau \cdot dA$$

We have also explicitly stated the definition of the Π indicator (eq. 3).

Moreover, we have added a detailed description of the method in the Supplementary Information following the suggestion of RC2.

**L151**: what assumptions?

The phrase "following our assumptions" referred to our definition of spice, but we agree with the reviewer that is confusing and not explanatory. We have clarified the definition of spice explicitly in the manuscript following Editor Comment L85 but also expanded in the definition of the terms and the methods of the water mass transformation description in the Supplementary information. We have then removed "following our assumptions" and just left "diaspice".

**L162**: Do you mean that you used ERA5 temp/salt data to find isotherms/isohalines that outcropped and used those values to identify the classes on your sigma-tau plot? If so, can you say that to be clear?

Thank you for the opportunity to clarify this. We did not use ERA5 temperature and salinity to identify outcropping classes directly. Instead, we applied the water-mass transformation (WMT) method in T-S space including transformation via air-sea fluxes. We used T-S observations from gliders and Argo, and ERA5 air-sea buoyancy fluxes, to determine which T-S classes experience surface buoyancy-driven transformation (following Evans et al., 2014, 2023). We then converted those transformed T-S classes that are affected by surface

buoyancy-driven transformation into σ-τ space and compared them with the σ-τ domain of the mode water. As shown in Fig. S2, the σ-τ classes affected by buoyancy fluxes lie well above the mode-water σ-τ range, confirming that air-sea fluxes do not directly influence the mode-water classes during our analysis period. We have modified the manuscript to explain this more clearly:

L175: "*To assess the role of surface forcing, we applied the water mass transformation framework in temperature-salinity (T-S) space (Evans et al., 2014; 2023). Using glider and Argo T-S observations, together with ERA5 air-sea buoyancy fluxes (Hersbach et al., 2020), we diagnosed the surface transformation of distinct T-S classes, thereby identifying which classes are actively transformed by air-sea fluxes. These classes were mapped into σ-τ space and compared with the σ-τ domain of the mode water (Figure S2), showing no overlap. The classes influenced by surface fluxes lie well above the density range of the mode water (Figure S2), indicating that surface buoyancy forcing does not influence the observed mode water transformations*"

**L184**: There are no panels for Figure 1 (f) and (g).

Thank you. Apologies for the mistake, those panels existed in a previous version of figure 1. Changed to Fig 1e.

**L185**: Reference Figure 2 in this sentence.

Added.

**L195**: cite please

"The previous analysis..." does not referee to an independent study. To remove the ambiguity, we start the sentence as "Over shorter timescales, the..." in L224.

**L200**: typos - 2f; "transf."

We have changed all of the "tranf" to "transf."

**Fig2**: (a) and (b) order should be switched in this figure along with the corresponding changes in text. **Fig2e**: "Denser" "lighter" should be inside the panel, it is visually busy/confusing the way it is currently placed. Same for "spicier" "mintier" **Fig 2f**: Make sure the colors chosen for the lines are accessible to readers with color vision deficiencies

Thank you. We have applied the changes you suggested in Figure 2 (now Fig 3). We haven't put the denser and lighter inside the panel, but put them closer so they don't feel that visually detached from panel e. We changed the line colors of figure f and accordingly changed the rest of the figures where diapycnal and isopycnal transformations are plotted.

[Figure]

**L220**: Can you explain why you integrated over spice class for isopycnal transformation and potential density for diapycnal transformation? This goes back to my comment in L60

Thank you for this comment. In the σ-τ framework, isopycnal and diapycnal transformations represent fluxes across τ and σ surfaces, respectively. Isopycnal transformation ($U\tau$) quantifies mixing along density surfaces, i.e., the redistribution of water masses within the same σ but across different τ. Because this mechanism acts horizontally in density-spiciness space, the natural way to express a bulk transformation is to integrate over τ within the mode water density band. Diapycnal transformation ($U\sigma$) quantifies mixing across density surfaces, i.e., vertical exchanges that move water into lighter or denser σ classes. This mechanism acts vertically in density-spiciness space, so the appropriate bulk representation is an integral over σ within the mode water τ range. This approach follows the standard interpretation of σ-τ transformations described in Evans et al. (2014) and Portela et al. (2020b), where integrating along the "inactive" coordinate isolates the component of the transformation driven by fluxes across the "active" coordinate. It also ensures that the resulting integrated values reflect the net tendency acting within the full mode water layer, rather than focusing on any single σ-τ bin.

**L234-241:** Very cool calculation to justify high sampling frequency!

Thank you.

**L246**: Figure 1a used gray for argo climatology and orange for glider - i suggest flipping the colors here (or in fig 1) to be consistent with the colors representing which dataset

Thank you. We have applied the changes you suggested in Figure 1 (orange for climatology and black for glider). See RC1 - Fig1.

**L283**: spell it out since this is the first mentioning in captions

Done.

**L284**: The small yellow diamond is hard to see, can you choose a different color (like, cyan or hot pink).

Thank you. We have changed the color of the diamond to black with a white edge for consistency with Figure 1 and 4 (previous Fig 3). Moreover, we removed the MLD for simplicity and changed the color and linestyle of the MW boundaries following Figure 1 to white dotted. Finally, we changed the color of the transformations to be consistent and colorblind friendly. We have changed the figure caption and the text accordingly.

[Figure]

**L285**: typo

Corrected.

**L295**: Paragraph explanation of fig 5 should come before the referencing of fig 5. Before L278.

Thank you for the suggestion. To improve clarity and maintain a consistent narrative flow, we removed the early reference to Figure 5 rather than relocating it.

**Fig6**: Perhaps this figure would be helpful to the reader before the other figs. consider placing this schematic as your fig 1 or 2.

Thank you for the suggestion. We have moved Fig. 6 to Fig. 2 and accordingly edited the text and all figure numbers. We used it in the description in Section 2.2:

L 147: *"A diagram illustrating how changes in water characteristics are represented in σ-τ space is shown in Figure 2a. The processes that modify the volume of a σ-τ class are depicted in geographical coordinates in Figures 2b-c. The σ-τ class highlighted with a square in Figure 1a is marked with dots in Figures 2b-c."*

---

## Author Comment (AC4)

We thank the editor for critically reading this manuscript and providing helpful feedback, which has added a great deal to improve the manuscript and clarify the text. We respond to all issues addressed in their comments below, as well as adding the revised changes in the manuscript. The Editor comments are included here in black, and our answers below their respective comments in blue. The text that has been modified in the manuscript according to the reviews is presented in *italic*. The line numbers in the answers refer to the marked-up manuscript version with tracked changes

**EC**

Dear authors,

The three reviewers were very positive. They provide several points for the authors to consider in their revised manuscript. In addition, I added my own comments below after reading of the manuscript. I'm also positive about this manuscript. I encourage the authors to consider all the comments and provide a point by point response to authors comments and revised manuscript to Ocean Sciences.

**L85** - It is written as Conservative Temperature and Absolute Salinity. Both are capitalized. Please also refer to the right papers that define these variables. For Conservative Temperature this is McDougall 2003, Graham and McDougall2013. For Absolute Salinity McDougall et al 2011.

- McDougall, T. J.: Potential Enthalpy: A Conservative Oceanic Variable for Evaluating Heat Content and Heat Fluxes., Journal of Physical Oceanography, 33, 945-963, https://doi.org/10.1175/1520-0485(2003)033<0945:PEACOV>2.0.CO;2, 2003.

- Graham, F. S. and McDougall, T. J.: Quantifying the Nonconservative Production of Conservative Temperature, Potential Temperature, and Entropy., Journal of Physical Oceanography, 43, 838-862, https://doi.org/10.1175/JPO-D-11-0188.1, 2013.

- McDougall, T. J., Jackett, D. R., Millero, F. J., Pawlowicz, R., and Barker, P. M.: A global algorithm for estimating Absolute Salinity., Ocean Science, 8, 1117-1128, 2012.

Thank you, we fixed the capital letters and also cited these references.

**L85** - Please provide a definition/equation for spice.

We use the definition from McDougall and Krzysik (2015), and we implemented it through the TEOS-10 routine *spiciness0*. The text in the manuscript has been modified to explicitly reference the function on the TEOS-10 routine:

*Line 86. "Spice ($\tau$; kg m$^{-3}$) was computed from Conservative Temperature (Graham and McDougall, 2013; McDougall, 2003) and Absolute Salinity (McDougall et al., 2012) following the TEOS-10 routines (gsw_spiciness0; McDougall and Barker, 2011). Potential density and spice are referenced at 0 dbar."*

*and in L132: "....spice is interpreted as a measure of thermohaline variability along isopycnals, reflecting isopycnally-compensated temperature and salinity changes associated with the spreading of distinct water masses (Jackett and McDougall, 1985; McDougall and Krzysik, 2015). "*

**L87** - Please define details about calculating EKE.

Thank you. We have defined and clarified EKE calculations. See comment L87-RC1.

**L95** - The current information is not enough for proper reproducing the data. Some questions I was left with are for example: You select all Argo floats within hundreds of km, as given by fig 1a? Then you take the gray line as center and bin selected Argo floats and take the median. But this selection, is this from a circle with a radius from the center? Or is this some lines perpendicular to the transect? Is this averaged on pressure surfaces or on isopycnal surfaces, and what are the consequences of this choice. What does the 3km horizontal scale have to do with this, as it is projected on a line?

We thank the Editor for highlighting the need for clearer methodological detail. We have now substantially expanded the description of how Argo profiles were selected, projected, and remapped onto the across-Gulf transect following your suggestion and RC1 and RC3. The 3 km horizontal grid is to have the same grid for all months to be able to perform the WMT analysis. The revised text appears in Section 2.1.

L101: *"Argo float coverage spans the entire domain, with an average density of 28 profiles per 0.25° × 0.25° grid cell (Figure 1a) and a relatively uniform monthly distribution (Figure 1c). Argo profiles within a 200 km distance from the across-Sea of Oman transect (Figure 1a, orange dashed line) were selected. This strategy ensures sufficient monthly sampling coverage in this sparsely observed region to construct an across-gulf monthly climatology. Moreover, to avoid the influence of shallow profiles on the continental shelf, profiles shallower than 1000 m were excluded, so the transect remains representative of the environment where mode water forms and persists. Each profile was then orthogonally projected onto the nearest point along the transect (the orange dashed line in Figure 1a), which provides its along-transect coordinate. Profiles were vertically interpolated onto a uniform 2-m pressure grid, and all projected profiles were median-binned into 3-km horizontal bins along the transect. Averaging was performed on pressure levels. Monthly climatologies were produced by taking the median across all profiles within each (depth, distance) bin for each month between 2000 and 2023. This gridded product is then used as input for the σ-τ water-mass transformation calculations."*

**L155** - The description requires a bit more explanation. For example, what are the coefficients in A? They can't be derived from the given information here. Perhaps provide a short appendix that the reader can refer to for a summary of the method. How sensitive are your results to choices within the inverse machinery (see Groeskamp et al 2017 as an example).

- Groeskamp, S., Sloyan, B. M., Zika, J. D., and McDougall, T. J.: Mixing Inferred from an Ocean Climatology and Surface Fluxes, Journal of Physical Oceanography, 47, 667-687, https://doi.org/10.1175/JPO-D-16-0125.1, 2017.

Thank you for this suggestion. We agree that the method needs more description (as also suggested by Reviewer 2). We thus have added an extended explanation addressing these concerns and provided a step-by-step description in the Supplementary Information. Please see RC2 - WMTF description.

We are currently uncertain about how best to apply the sensitivity of the inverse framework used in our water mass transformation (WMT) analysis. Our approach uses a simplified formulation of the WMT framework compared to Groeskamp et al. (2017), in which we include only mixing contributions to the transformation. Specifically, we adapted the code from Evans et al. (2014) to our dataset, which we made available on GitHub. This implementation constructs the matrix A and solves the linear system using a least-squares approach, without applying additional weighting or preconditioning. We are therefore unclear

on how to adapt this simplified inverse setup to directly address the sensitivity issue raised in your comment. We feel that the manuscript is sufficient without including a sensitivity analysis at this point but we are open to discussing the issue further if you could provide further clarification on an approach applicable to our implementation.

**L170** - How do you define a surface area from a transect? This surface area is needed in eq 1.1 and 1.2.

A hydrographic transect is 2-D (distance x depth). Each measurement can be assigned to a σ-τ class. For each class, you look at the portion of the transect cross-section occupied by water in that class. This gives you an area in units of $m^2$, not $m^3$ or $m^2$ per meter of width. That area is what enters your isopycnal/isospice integrals.

We have rephrased the definition in L159: *"...and dA is the cross-sectional area of the transect occupied by water within the specified σ-τ class. .."* and added details on the calculation in the Supplementary Information 1 following Evans et al., 2014.

**Section 3.1 and figure 2:** Spice is not defined. WMT is in units of m2/s which is a flux. It is unclear from the text or equations how these units come to be and why it is not kg/s or m3/s.

Following your previous comment, now Spice has been defined. There is also an explanation of the units of the transformation fluxes in the new detailed Supplementary Material: *"These dia-surface transformations should be interpreted as volume fluxes of water and have units of m3 s-1. They cannot be practically diagnosed from velocity measurements and must therefore be determined indirectly from changes in the volumetric distribution of water projected into σ-τ coordinates. In the case of a two-dimensional ocean transect, as per this study, the method is identical; however, the inferred transformations are area fluxes and have units of m2 s-1."*

**Fig 2e** shows potential density anomaly. Please clarify the following things: 1) Sigma is already a symbol for potential density anomaly (1000 is subtracted, see TEOS-10). 2) please provide equations defining spice anomaly and potential density anomaly. Please clarify this statement to make it clear how anomalies are calculated: "the glider after applying a 10-day rolling mean (solid), and the climatology (dotted)

We apologies for the confusion. In this study, σ refers to potential density ($σ = ρ - 1000$ kg $m^{-3}$), following TEOS-10 conventions. To avoid confusion, we explicitly define potential-density and spice temporal anomalies as: $σ'(t)=σ(t)-\barσ$ and $τ'(t)=τ(t)-\barτ$ where $\barσ$ and $\barτ$ are the time-means computed over the analysis period (mid-March to July). These temporal anomalies quantify deviations from the mean state of the mode water layer.

We have changed the description in the caption of Figure 2e (now 3e) accordingly to clarify: *"Temporal anomalies of potential density ($σ'$) and spice ($τ'$) computed as deviations from their time-mean over the March–July period as $σ'(t)=σ(t)-σ$ and $τ'(t)=τ(t)-τ$ where $σ$ and $τ$ are the time-means computed over the analysis period. Solid light lines show the 3-day glider anomalies; solid dark lines show the glider anomalies after applying a 10-day rolling mean; dotted lines show the monthly climatological anomalies. "*

How does cabbeling and thermobaricity affect density changes in this method?

Cabbeling and thermobaricity are accounted for implicitly. Because all σ-τ calculations use TEOS-10 CT, SA, and $σ_0$, the nonlinear equation-of-state effects that arise from cabbeling and thermobaricity are already resolved in the density tendencies. The WMT framework interprets any resulting cross-density-surface flux as part of the diapycnal transformation.

Thermobaric effects are expected to be minimal in our domain (upper 500 m, $\sigma_0$), but cabbeling may contribute to the diagnosed diapycnal term; however, the method does not allow isolating this from turbulent diffusivity.

**L226-245**. This is an interesting analysis and relevant. However, I have a lot of trouble understanding exactly what is done, from the explanation given. Its probabaly all there, but I would encourage the authors to find a clearer way of explaining and presenting these results.

Thank you for this comment. We acknowledge that this section presents a more technical analysis, and we carefully re-evaluated the text for clarity. We have opted not to restructure the section but have made several targeted edits to improve readability:

L262: "*..This approach provides a distribution of possible means for each effective sampling resolution. To illustrate how smoothing influences variability, we also applied rolling means of increasing window length to the 3-day series (shown as violin plots in Figure 4). As the window size increases, extreme values in both isopycnal and diapycnal transformation are progressively damped (Figure 4).*".

Moreover, we changed the previously defined *"true mean"* as *"3-day mean" as it is more explicit and clear (following RC2).*

Please repeat the meaning of ADT in caption of figure 4

Done.

**L375** - in this paragraph, it is said that climatological WMT will miss peaks. However, to what extend is this related to the method applied here? This is not a statement that can easily be broadened to all WMT as these rely on different approached to do the actual calculations. Maybe adapt this paragraph to the specific method used. Be careful or specific about the broader statements.

The comment refers to applications to climatological data which by nature of its delta(time) cannot resolve sub-monthly variability and will instead present a time-integrated result. This independent of WMTF method choice. What we meant was that monthly averages will miss higher frequency variability, therefore we have modified the text and referenced the figures supporting our statements:

L405: "As a result, climatological approaches miss sub-monthly variability and underestimate both the intensity and variability of transformation processes (Figure 4), particularly the contributions from isopycnal stirring and advective exchange (Figure 6)."

**L390** - How would microstructure provide lateral mixing estimates?

We thank the reviewer for putting attention into this statement. The original phrase was *"..Including turbulence measurements, such as microstructure-derived diffusivities, would help disentangle the relative roles of vertical mixing, lateral stirring, and advection...".* We agree that it was confusing as per the enumeration of processes after microstructure observations. With this sentence we wanted to emphasize that the WMTF provides an indirect measurement of diapycnal mixing, but this vertical mixing could be highly influenced by transient processes like lateral intrusions that could enhance/supress gradients, and due to the intrinsic way to compute the WMTF we can not capture this variability. Moreover, this region is characterized by double diffusive convection instability (Font et al., 2024) and has been shown that this process can enhance more than 50% the local diapycnal mixing diffusivity (Fischer et al 2013; Pinto-Juica et al. *accepted for Nat. Comms. E&E*), playing an important role in the oxygen

redistribution below the MLD. Therefore, we want to be explicit that there is an important contribution that we can not capture due to methodological constraints, but we know it's important. We have rephrased the statement to remove the ambiguity to:

*L417: "...Including turbulence measurements, such as microstructure-derived diffusivities to resolve the role of vertical mixing, could allow for disentangling those from lateral stirring, and advection, and thus refine the interpretation of transformation processes in dynamic regions like the Sea of Oman."*

Fischer, T., Banyte, D., Brandt, P., Dengler, M., Krahmann, G., Tanhua, T., and Visbeck, M.: Diapycnal oxygen supply to the tropical North Atlantic oxygen minimum zone, Biogeosciences, 10, 5079–5093, https://doi.org/10.5194/bg-10-5079-2013, 2013.

Juica, M., Pizarro, O, Santana, A., Valencia, L., Ulloa, O., Figueroa, P., Ramos, M. & Queste, B.: Salt Fingers Contribute Substantially to Diapycnal Oxygen Transport into the Oxygen Minimum Zone of the Eastern South Pacific. 10.21203/rs.3.rs-7151709/v1. (2025 - accepted).

I think it would help the manuscript if Figure 6 becomes figure 1.

Thank you for the suggestion. We have moved Fig. 6 to Fig. 2 and accordingly edited the text (see RC3).

---

## Editor Decision (ED1)

**Second review: Spatio-temporal scales of mode water transformation in the Sea of Oman**

**Josef Bisits**

**January 12, 2026**

In the revised manuscript, the authors have clearly responded to all of my suggestions. I believe the manuscript is ready for acceptance. I do offer a few very minor comments relating to the material added only to offer improved readability.

**Minor comments**

**Line 155:** the word 'then' is not required on this line.

**Line 160:** The updated mathematical description of $\Pi$ is good. I think to help a reader even more, a short description of what $\Pi$ is in words would be a good addition to the text here e.g. "The function $\Pi$, defined as ..., determines if a water parcel falls into a chosen $\sigma - \tau$ class."

**Line 165-166:** I suggest changing 'built' to 'generated', 'link' to 'equate', 'to' to 'and', and removing 'as' prior to the inline volume trend equation. So the sentence becomes: *Using equation [1], a set of linear equations can be generated to equate the volume trend and the interior water-mass transformation in $\sigma - \tau$ coordinates,* $\mathrm{d}\mathbf{V}/\mathrm{d}t = \mathbf{A}\mathbf{x}$... I appreciate this is the study of the authors so if they do not find this suggestion helpful, and the current wording clear enough, they may disregard this suggestion.

**Supplementary info:** Remove the bold font from the word 'is' in the dot point describing the vector of unknowns $\mathbf{x}$.

**Supplementary info:** On the fourth last line, the $\sigma$ and $\tau$ should be subscripts to the $U$.

**Supplementary info:** In the last sentence, I suggest changing 'The detailed methodology' to something like "See also Evans et al (2014) and Portela et al (2020) for other explanations of the methodology." The authors have provided plenty of detail in this supplementary info!

---

## Author Response (AR2)

We thank the referee for providing helpful feedback. We respond to all issues addressed in their comments below, as well as adding the revised changes in the manuscript. The Reviewer comments are included here in black, and our answers below their respective comments in blue.

In the revised manuscript, the authors have clearly responded to all of my suggestions. I believe the manuscript is ready for acceptance. I do offer a few very minor comments relating to the material added only to offer improved readability.

- Minor comments Line 155: the word 'then' is not required on this line.
  Removed.
- Line 160: The updated mathematical description of Π is good. I think to help a reader even more, a short description of what Π is in words would be a good addition to the text here e.g. "The function Π, defined as ..., determines if a water parcel falls into a chosen $\sigma - \tau$ class."
  Changed.
- Line 165-166: I suggest changing 'built' to 'generated', 'link' to 'equate', 'to' to 'and', and removing 'as' prior to the inline volume trend equation. So the sentence becomes: Using equation [1], a set of linear equations can be generated to equate the volume trend and the interior water-mass transformation in $\sigma - \tau$ coordinates, $dV/dt = Ax$... I appreciate this is the study of the authors so if they do not find this suggestion helpful, and the current wording clear enough, they may disregard this suggestion.
  Changed as suggested.
- Supplementary info: Remove the bold font from the word 'is' in the dot point describing the vector of unknowns x.
  Fixed.
- Supplementary info: On the fourth last line, the $\sigma$ and $\tau$ should be subscripts to the U.
- Fixed.
- Supplementary info: In the last sentence, I suggest changing 'The detailed methodology' to something like "See also Evans et al (2014) and Portela et al (2020) for other explanations of the methodology." The authors have provided plenty of detail in this supplementary info.
  Changed as suggested.